# PROT2TOKEN: A UNIFIED FRAMEWORK FOR PROTEIN MODELING VIA NEXT-TOKEN PREDICTION

## ABSTRACT

The diverse nature of protein prediction tasks has traditionally necessitated specialized models, hindering the development of broadly applicable and computationally efficient Protein Language Models (PLMs). In this work, we introduce Prot2Token, a unified framework that overcomes these challenges by converting a wide spectrum of protein-related predictions—from sequence-level properties and residue-specific attributes to complex inter-protein interactions—into a standardized next-token prediction format. At its core, Prot2Token employs an autoregressive decoder, conditioned on embeddings from pre-trained protein encoders and guided by learnable `task tokens`, to perform diverse predictions. This architecture uniquely facilitates multi-task learning, enabling general-purpose decoders to generalize across five distinct categories. We present extensive experimental validation across a variety of benchmarks, demonstrating Prot2Token's predictive power in different types of protein-prediction tasks. In 3D structure prediction, Prot2Token delivers substantial speedups (up to ∼1000× faster than AlphaFold2 with MSA on the same hardware) while, across other numerous tasks, matching or surpassing specialized methods. Beyond that, we introduce an auxiliary self-supervised decoder pre-training approach to improve spatially sensitive task performance. Prot2Token thus offers a step towards standardizing biological prediction into a generative interface, promising to accelerate biological discovery and the development of novel therapeutics.

## 1 INTRODUCTION

Proteins are the fundamental building blocks of life, playing a critical role in maintaining human health. However, understanding the complex language of proteins—encoded in their sequences and structures—remains a significant challenge for researchers (Shim et al., 2019). This complexity limits our ability to interpret, predict, and design proteins for various biomedical and therapeutic applications.

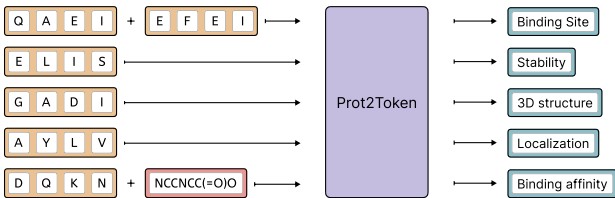

Figure 1: High-level architecture of *Prot2Token* highlighting multi-task capability in protein-level, residue-level, and protein-protein level tasks.

Protein function prediction is particularly challenging due to the vast diversity of protein sequences, structural variations, and the limited availability of labeled data. Unlike natural languages, protein sequences do not follow explicit syntactic rules understandable by humans, making it difficult for models to learn meaningful representations without extensive biological knowledge (Ofer et al., 2021). Protein language models (PLMs) offer a transformative solution by learning meaningful representations of protein sequences, enabling researchers to decode and translate protein data into a more interpretable format (An & Weng, 2022; Ferruz & Höcker, 2022). By leveraging PLMs, we can

bridge the gap between raw protein information and human understanding, advancing research in drug discovery, disease mechanisms, and synthetic biology.

While PLMs have significantly advanced protein-prediction tasks, current models require task-specific specialization after pre-training (Hu et al., 2023; Roche et al., 2024). This reliance on separate modules for distinct tasks leads to inefficient computational resource use and limited scalability. Most PLMs undergo post-training alignment with specialized predictor architectures for individual tasks, requiring independent training and fine-tuning—a time-consuming and resource-intensive approach (Weissenow & Rost, 2025). A unified tokenization protocol capable of standardizing diverse protein-prediction tasks would overcome this limitation, streamlining protein function prediction and enhancing its accessibility for real-world applications.

To the best of our knowledge, despite the emergence of foundation models for proteins, no comprehensive strategy exists to systematically map these heterogeneous outputs into a shared generative space. Instead, researchers often modify existing foundation models to suit particular applications (Schmirler et al., 2024), such as predicting 3D protein structures from sequences using customized techniques (Jumper et al., 2021; Lin et al., 2022). One key limitation is that most existing models are based on BERT-style architectures (Unsal et al., 2022), while effective for providing meaningful representation, lack the flexibility needed for diverse and controllable prediction capabilities. In natural language processing (NLP), the transition from BERT-style models to autoregressive GPT-style models has enabled more dynamic and human-understandable instructions (prompts) to control the generation process and therefore, handling a diverse set of predictions within the NLP domain (Ouyang et al., 2022; Achiam et al., 2023). A similar paradigm shift is necessary in protein research, moving beyond static encoders toward more advanced generative AI approaches that provide more comprehensive predictive capabilities.

Although autoregressive transformer models have been explored for the language of protein—such as ProGen2 (Nijkamp et al., 2023), RITA (Hesslow et al., 2022), and Ankh (Elnaggar et al., 2023)—they struggle with controllability and task, especially for protein-prediction tasks. Unlike language models in NLP, which effectively leverage prompting mechanisms for controllable and interpretable predictions, autoregressive PLMs currently lack robust methods to guide their outputs toward human-interpretable formats. This gap hinders their practical applicability and, in contrast to NLP, has compelled researchers to continue relying heavily on encoder-style PLMs, often building specialized architectures around these encoders for specific protein prediction tasks.

To address these limitations, this work takes a significant step toward unifying protein prediction by establishing a universal tokenization protocol that categorizes diverse tasks into five sets. We introduce a universal protocol for tokenizing different protein-prediction tasks, enabling a general autoregressive transformer predictor to leverage existing BERT-style PLMs (Figure 1). This generative approach, guided by a unified next-token prediction loss, demonstrates generality across multiple protein-prediction task categories, including protein-level, residue-level, and protein-protein interaction-level tasks. We illustrate its versatility through extensive evaluation on five categories of tasks: Classification, Regression, Binding Site, Sequence-to-Sequence, and Other. Specific examples evaluated include kinase phosphorylation site prediction, protein-ligand binding site prediction, protein 3D structure prediction, and protein mutation stability assessment. Furthermore, our framework inherently supports multi-task learning, and we provide initial analyses demonstrating synergistic performance improvements when related tasks are trained jointly.

For certain specialized tasks, such as predicting binding sites, we show that initializing the decoder through self-supervised pre-training significantly boosts performance. Specifically, for protein-ligand binding site prediction, we further analyzed the learned token representations, revealing meaningful relationships among ligand tokens that enabled us to enhance predictions for underrepresented ligands. We believe that our approach represents an essential step toward harnessing and upgrading large language models (LLMs) for robust and flexible protein prediction tasks.

## 1.1 RELATED WORK

Many specialized or *foundation* models now exist for proteins (Wang et al., 2025b), yet none provides a single, prompt-controllable interface capable of both generation and a diverse set of prediction tasks. We therefore group prior works into generative protein design, predictive representation learning, and unified models.

**Generative protein design.** Autoregressive language models dominate *de novo* sequence generation. *ProGen* first demonstrated controllable generation using functional tags (Madani et al., 2020). Subsequent scaling—*ProtGPT2 1.2b* (Ferruz et al., 2022), *RITA 1.2b* (Hesslow et al., 2022), and *ProGen2 6.4b* (Nijkamp et al., 2023)—improved perplexity and experimental success yet still require task-specific fine-tuning or filtering to steer functions. Most recently, *ProGen3* extends this trend by scaling it up significantly, but reports limited controllability for fine-grained generation (Bhatnagar et al., 2025).

**Predictive representation learning.** A parallel thread focuses on bidirectional encoders that power task-specific heads. Large masked-language models such as *ESM2-15b* yield embeddings for a spectrum of downstream tasks (Lin et al., 2022) and even drive end-to-end folding with *ESMFold* (Lin et al., 2022)—yet the folding module is specialized for 3-D structure prediction. Likewise, *AlphaFold2* (AF2) couples *EvoFormer* encoders to a bespoke structure decoder (Varadi et al., 2022). Such "wrapper" architectures excel at their dedicated outputs but do not form a general predictor. We find only one cross-task autoregressive alternative: *PTMGPT2* (Shrestha et al., 2024), which adapts *GPT-2* with prompt-based fine-tuning to predict 19 classes of post-translational modifications (PTMs) in a single model—still restricted to the PTMs domain.

**Unified models.** Recently, models have emerged that aim to link protein design and prediction within a single system. *HelixProtX* unifies sequence, structure, and free text in one multimodal autoregressive transformer, capable of translating between any two of those modalities and predicting atom-level 3-D structure directly from sequence (Chen et al., 2024). *ProLLaMA* (Lv et al., 2024) adapts *LLaMA-2* through protein-specific instruction tuning so that one model, guided by natural-language prompts, can perform controllable sequence generation together with property-prediction tasks such as stability, fluorescence, binding affinity, and remote-homology classification (Lv et al., 2024). *InstructProtein* aligns protein sequences with human language via knowledge-graph–guided instruction tuning, allowing the model either to describe a protein's function in free text or to generate a plausible sequence that satisfies a textual specification (Wang et al., 2023). Although these systems demonstrate encouraging modality transfer, they still depend on prompt engineering for fine-grained control and have yet to be benchmarked across the full suite of standard prediction tasks addressed in this work.

## 2 METHOD

### 2.1 PROT2TOKEN ARCHITECTURE

The *Prot2Token* framework is designed to unify diverse protein-related prediction tasks using a shared architecture based on encoder-decoder transformers. The core idea is to integrate an autoregressive decoder language model with existing encoder-style protein and optional chemical language models via cross-attention layers, thereby converting prediction tasks into a unified next-token prediction problem.

The architecture employs a pre-trained bidirectional transformer (*ESM2*) as the protein encoder. For tasks involving chemical information (e.g., ligand binding), an optional chemical encoder (*BARTSmiles*) (Chilingaryan et al., 2022) is used to process *SMILES* representations. These encoders transform their respective input sequences into contextual embeddings:

$$h_{\text{enc}} = f_{\text{enc}}(x)$$

where $h_{\text{enc}} \in \mathbb{R}^{N \times d_{\text{enc}}}$ is the encoder output, $N$ is the sequence length, and $d_{\text{enc}}$ is the encoder's hidden dimension.

We use distinct embedding tables for each encoder (protein and, if applicable, chemical) and the decoder to reflect their differing tokenization schemes and functional roles in the architecture.

To enhance the position-awareness of the sequence embeddings, we introduce a learnable positional embedding layer $g_{\text{pos}}(\cdot)$, producing augmented representations:

$$h_{\text{aug}} = h_{\text{enc}} + g_{\text{pos}}(p)$$

where $p \in \mathbb{R}^{N \times d_{\text{enc}}}$ is the learnable positional embedding.

To align the encoder output with the decoder's hidden dimension $d_{\text{dec}}$, we apply a linear projection:

$$h_{\text{proj}} = h_{\text{aug}} W_{\text{proj}} \quad \text{where } W_{\text{proj}} \in \mathbb{R}^{d_{\text{enc}} \times d_{\text{dec}}}$$

This projected representation $h_{\mathrm{proj}} \in \mathbb{R}^{N \times d_{\mathrm{dec}}}$ is fed into the decoder via cross-attention.

The decoder is a causal (autoregressive) transformer composed of standard transformer components such as multi-head self-attention, feed-forward layers, and GeLU activations. *FlashAttention-2* is incorporated to improve training speed and memory efficiency. For specific architectural configurations used in this work, refer to Table 10.

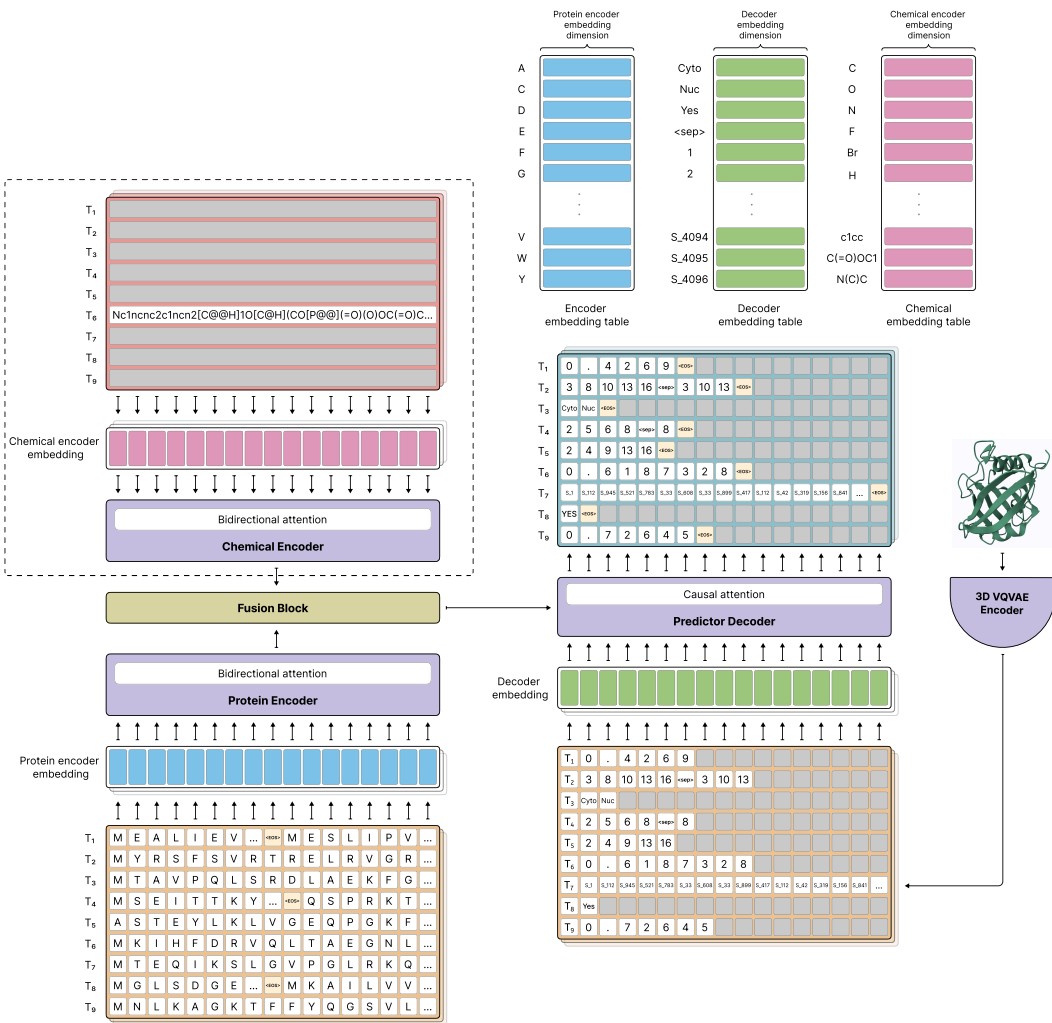

Figure 2: Detailed Architecture of *Prot2Token* Highlighting Multi-Task Capability. This diagram shows the *Prot2Token* components: a bidirectional Protein encoder and an optional Chemical Encoder, a Fusion block part, and an autoregressive Decoder guided by Task Token Embeddings for various prediction tasks (examples listed). This illustrates the framework's potential for simultaneous multi-task learning; however, practical training of this work only focused on combinations of fewer tasks due to computational costs, demonstrating the principle.

To support multiple tasks within a unified training process, we introduce `task token`. These tokens, placed at the beginning of each output sequence, serve as prompts that guide the decoder's behavior for each specific task. The task token sequence $t = (T_1, T_2, \ldots, T_m)$ is embedded via a learnable embedding function:

$$e_{\mathrm{task}} = g_{\mathrm{task}}(t) \in \mathbb{R}^{m \times d_{\mathrm{dec}}}$$

The decoder receives the embedded task tokens and attends to both them and the projected encoder outputs:

$$y = f_{\mathrm{dec}}(h_{\mathrm{proj}}, e_{\mathrm{task}})$$

During inference, the decoder is autoregressive: it receives a special beginning-of-sequence (<BOS>) token followed by the task token, and generates each output token sequentially.

The decoder factorizes the probability of the output sequence $x = (x_1, x_2, \ldots, x_T)$ as:

$$p(x) = \prod_{t=1}^{T} p_\theta(x_t \mid x_1, \ldots, x_{t-1})$$

The training objective is to minimize the negative log-likelihood:

$$L(\theta) = -\sum_{t=1}^{T} \log p_\theta(x_t \mid x_1, \ldots, x_{t-1})$$

To better manage the role of prompt tokens, we assign token-specific weights $w_t \in [0, \infty)$ to control their contribution to the loss. Specifically, we set $w_1 = 0$ to exclude the prompt (task token) from the loss, while allowing other tokens $t \geq 2$ to be weighted differently:

$$L(\theta) = -\sum_{t=1}^{T} w_t \log p_\theta(x_t \mid x_1, \ldots, x_{t-1})$$

This flexible weighting helps tune the model's attention to different parts of the label sequence.

Refer to Figure 2 for an overview of the *Prot2Token* architecture and Figure 4 for a closer look at how task tokens interact with the decoder. Architectural variants and configuration details are summarized in Table 10. By representing diverse outputs as token sequences, this design allows *Prot2Token* to unify a broad spectrum of protein prediction tasks under a single decoder, facilitating both joint and independent training regimes.

## 2.2 TOKENIZATION

The *Prot2Token* framework utilizes distinct tokenization strategies for its input encoders and the output decoder. Input sequences, such as protein amino acid sequences or chemical *SMILES* strings, are processed by the native tokenizers of their respective pre-trained encoders (e.g., *ESM2* for proteins, *BARTSmiles* (Chilingaryan et al., 2022) for chemicals). The core innovation resides in the unified tokenization strategy for the output labels predicted by the autoregressive decoder. This strategy is pivotal as it converts a wide array of biological prediction targets into standardized sequences of discrete tokens, enabling the decoder to handle diverse tasks via a consistent next-token prediction mechanism. All tokenized output sequences commence with a <BOS> token and conclude with an <EOS> token, clearly demarcating sequence boundaries.

As depicted in Figure 3, this approach transforms heterogeneous labels into a uniform sequential format, facilitating a task-agnostic decoding process. Specifically, for classification tasks, labels are mapped to unique discrete tokens, with multi-label tasks typically concatenating these tokens (often alphabetically). Regression tasks represent continuous numerical values through a granular digit-by-digit encoding of their character components (e.g., sign, digits, decimal point). Sequence-to-sequence tasks generate an output token for each residue in the input protein, maintaining a direct correspondence. Binding site prediction involves tokenizing the sorted 1-based indices of residues participating in interactions. Other complex output types, such as for PTMs, are also converted into specific token sequences, for instance, by listing potential and confirmed modification sites separated by a special <SEP> token. This universal tokenization protocol is fundamental to *Prot2Token*'s ability to unify a broad spectrum of protein prediction tasks within a single decoding architecture. Refer to Appendix A.2 for a comprehensive explanation of each specific tokenization method.

## 2.3 DATASETS

This work leverages a diverse set of tasks drawn from several established benchmarks and repositories, including PEER (Xu et al., 2022), ProteinShake (Kucera et al., 2023), CATH (Wang et al., 2025a), AlphaFoldDB (Varadi et al., 2022), and other curated sources such as ProteinGym (Notin et al., 2023). These datasets encompass a wide range of protein-related prediction tasks, including regression, classification, binding site, and sequence-to-sequence predictions. Details for each task, including preprocessing steps, are provided in Appendix A.3. All tasks in these datasets are tokenized according to the unified protocol described in Section 2.2.

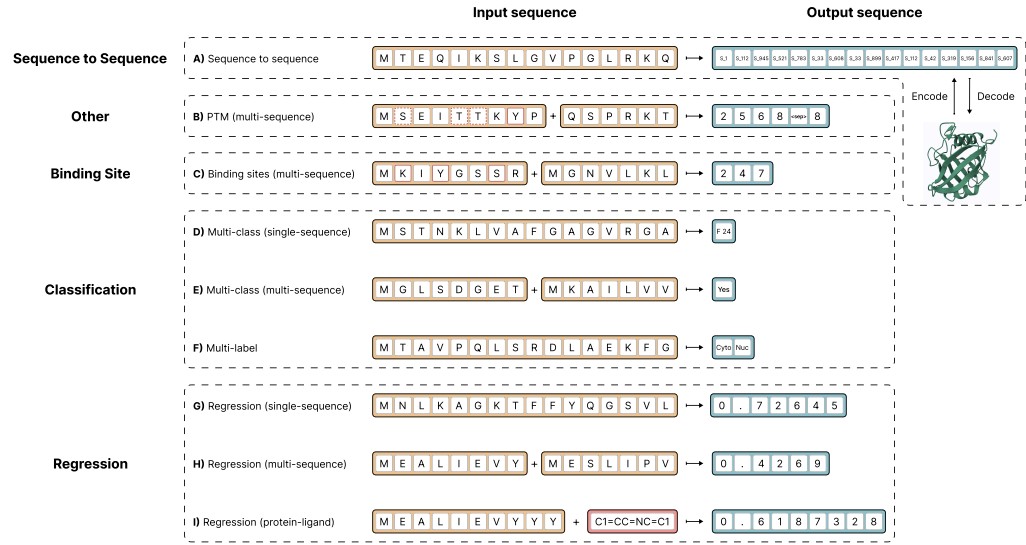

Figure 3: *Prot2Token* converts heterogeneous labels into uniform sequences: examples illustrate the five tokenization categories—(i) sequence-to-sequence, (ii) classification (multi-class/ multi-label), (iii) regression, (iv) binding-site indexing, and (v) other composite outputs such as PTM—highlighting the framework's task-agnostic decoding format.

## 3 EXPERIMENTS

We evaluated *Prot2Token* in multiple tasks on different datasets, including the protein-level, residue-level, and protein-protein level. For a subset of these tasks, we incorporated a self-supervised pre-training stage for the autoregressive decoder as an initial step. In all experiments, the protein encoder in *Prot2Token* was initialized using the pre-trained *ESM2-650m* model. For the decoder part, we used an autoregressive language model with different configurations based on the size of the *ESM* encoder and hyperparameters of the autoregressive decoder (Appendix A.1). We only considered *BARTSmiles* as the chemical encoder for the protein-ligand affinity task and disabled it for the other tasks. We optimized the number of unfrozen ESM-2 encoder layers for each task to align model capacity with task complexity and data availability; specific hyperparameters for each task are detailed in Appendix A.4.

Optimization was carried out with the AdamW optimizer (Loshchilov, 2017), applying a weight decay of 0.1 and using beta-1 and beta-2 values of 0.9 and 0.98, respectively, while setting epsilon to 1e-7. The learning rate followed a cosine annealing schedule with an initial warm-up phase (Loshchilov & Hutter, 2016), starting at 1e-6 and gradually increasing to 5e-5 over the first 256 steps unless stated otherwise. The training was performed using the PyTorch 2 framework (Ansel et al., 2024) on a single computational node equipped with four Nvidia A100 GPUs (80GB each).

### 3.1 CLASSIFICATION

This category includes multi-class, multi-label and hierarchical classification tasks such as Deeploc 2.0 and ER. The results are shown in Tables 1 and 2. In Deeploc 2 dataset, we significantly improved the performance compared to the original method, and also, the ER task result showed that the performance was boosted 7.5 percent by using multi-task learning. We could not calculate the Fmax metric for the EC and GO tasks, so we only considered the accuracy and $F_1$ scores to evaluate performance. Consequently, direct comparisons with other methods were not possible. Supplementary results with additional details are in Appendix A.4.1.

### 3.2 REGRESSION

This category encompasses four tasks: protein stability prediction, fluorescence intensity prediction, protein-ligand binding affinity estimation, and protein mutation stability assessment. The first two

tasks utilize a single protein sequence as input. In contrast, the protein-ligand affinity task takes both a protein sequence and a molecular *SMILES* string as input, while the mutation stability task uses a pair of protein sequences representing wild-type and mutant variants.

Table 1: Localization prediction using Deeploc-2 dataset. The results are based on the independent test set.

| Method | Macro-F$_1$ | Encoder |
|---|---|---|
| Baseline | 0.449 | ESM2-650m |
| Deeploc-2 (Thumuluri et al., 2022) | 0.46 | ProtT5 |
| Prot2Token-B | **0.5364** | ESM2-650m |

Table 2: Comparing methods on ER dataset. PLA and ST stand for protein-ligand affinity and stability, respectively. †: chemical encoder is attached.

| Method | Aux-Tasks | Accuracy | Encoder |
|---|---|---|---|
| Baseline | - | 83.81 | ESM2-650m |
| CoupleNet (Hu et al., 2023) | - | 89.0 | ProtT5 |
| Prot2Token-B | - | 79.29 | ESM2-650m |
| Prot2Token-B† | Deeploc+PLA+ST | **86.83** | ESM2-650m |

The results for these tasks are presented in Tables 3, 4, 6, and 7. Additional experimental details can be found in Appendix A.4.2. Across all regression tasks, *Prot2Token* consistently outperformed baseline methods from the PEER benchmark. Notably, in the fluorescence prediction task, multi-task learning led to a performance gain of up to 5.6% (Table 7). For mutation stability prediction, *Prot2Token* achieved a substantial improvement of over 51.5% compared to the best-performing baseline model as shown in Table 4.

## 3.3 BINDING SITE

We evaluated *Prot2Token* on two binding site prediction tasks: protein-ligand and protein-protein. For protein-ligand binding sites, each ligand type is represented by a dedicated task token in the decoder, which enables the model to capture ligand-specific interactions directly from protein sequences and learnable task tokens.

Table 3: Comparing protein-ligand affinity prediction methods on the test set. †: chemical encoder is attached.

| Method | RMSE | Encoder |
|---|---|---|
| PEER (Xu et al., 2022) (fine-tuned) | 1.559 | ESM1-1b |
| PEER (Xu et al., 2022) (fine-tuned) | 1.562 | ProtBert |
| Prot2Token-B† | **1.3887** | ESM2-650m |

Table 4: Comparison of mutation effect prediction models on the ProteinGym benchmark with original supervised 5-fold cross-validation indices. Additional baselines are included from the original ProteinGym paper (Notin et al., 2023). † denotes a linear layer fine-tuned on the last four encoder blocks.

| Method | Spearman |
|---|---|
| ESM-1v | 0.542 |
| MSAT | 0.568 |
| Tranception | 0.571 |
| ProteinNPT | 0.613 |
| Baseline (ESM-2†) | $0.8812 \pm 0.003$ |
| Prot2Token-C | **0.9294** $\pm$ 0.0018 |

Table 5: F$_1$ scores for the top 10 ligands across different training configurations on the test sets, with varying numbers of auxiliary ligands. The table summarizes the impact of jointly training with 10, 20, 30, and 41 ligands on binding site prediction. † indicates that self-supervised tasks were excluded during supervised training.

| Ligand | 10 ligands † | 10 ligands | 20 ligands | 30 ligands | 41 ligands |
|---|---|---|---|---|---|
| **Average** | 0.1883 | 0.6076 | 0.5942 | 0.6181 | 0.6132 |
| **Weighted Average** | 0.1849 | 0.6297 | 0.6277 | 0.6368 | 0.6353 |

We introduced a separate self-supervised pre-training stage for the decoder weights to enhance model initialization to improve predictive performance of binding site prediction-type tasks before training such tasks. This strategy significantly improves the model's ability across tasks require a wide range of binding site indices. A detailed rationale and methodology for this self-supervised pre-training are provided in Appendix A.4.3. We reported high-level performance results of protein-ligand binding site prediction in Table 5, demonstrating that *Prot2Token* achieves competitive predictive accuracy across various ligand types with the help of self-supervised pre-training (see detailed results of this task and protein-protein binding site in Appendix A.4.4).

Table 6: Comparing *Prot2Token* with other methods on stability prediction.

| Method | Spearman | Encoder |
|---|---|---|
| Baseline | 0.7527 | ESM2-650m |
| PEER(Xu et al., 2022) (fine-tuned) | 0.75 | ESM1-1b |
| PEER(Xu et al., 2022) (fine-tuned) | 0.771 | ProtBert |
| Prot2Token-B | **0.7947** | ESM2-650m |

Table 7: Comparing fluorescence prediction methods w/ and w/o multi-task learning. PLA and ST stand for protein-ligand affinity and stability, respectively. We considered the fine-tuned methods of PEER as the comparison. †: chemical encoder is attached.

| Method | Aux-tasks | Spearman | Encoder |
|---|---|---|---|
| Baseline | - | 0.676 | ESM2-650m |
| PEER (Xu et al., 2022) | - | 0.679 | ESM1-1b |
| PEER (Xu et al., 2022) | - | 0.679 | ProtBert |
| Prot2Token-B | - | 0.7389 | ESM2-650m |
| Prot2Token-B† | PLA | 0.7766 | ESM2-650m |
| Prot2Token-B† | PLA+ST | **0.78** | ESM2-650m |

Furthermore, to understand the representation learned by the task tokens, we explored their embeddings and identified relationships that correlate closely with biochemical properties (Appendix A.5). These findings, visualized in Appendix, Figure 17, indicate that task tokens not only serve as input identifiers but also encode biologically relevant information. Leveraging these insights, we further utilized the learned relationships to boost predictive accuracy for underrepresented ligands, achieving significant performance gains as summarized in Table 34. More details in Appendix A.5.5.

### 3.4 SEQUENCE-TO-SEQUENCE

In this part, we evaluated *Prot2Token* on residue-wise sequences by formulating it as a sequence labeling task, where the model generates a discrete token for each residue in the input protein sequence. The main focus of this section is on the challenging task of sequence-to-3D structure prediction. Here, *Prot2Token* is trained to generate discrete 3D structure tokens from amino acid sequences using a vector quantized variational autoencoder (VQ-VAE) based representation for protein backbone coordinates. The results are summarized in Table 8, which reports TM-score and runtime for representative structure prediction methods. Notably, *Prot2Token-D* demonstrates a dramatic speed advantage, producing structure predictions for a typical 384-residue protein in 1–2 seconds on a single A100 GPU—approximately three orders of magnitude faster than *AF2* with multiple sequence alignment (MSA) input, which typically requires 18–25 minutes for inference. This substantial speed-up makes *Prot2Token* particularly well-suited for large-scale or real-time structure generation scenarios. Representative examples of successful and unsuccessful 3D structure predictions are illustrated in Figure 13 and Figure 14, respectively. Interestingly, although the validation perplexity continues to decrease (Figure 15), structure accuracy plateaus at ∼0.55 TM-score on CAMEO 2024; this aligns with the ∼0.60 TM-score reconstruction ceiling of the VQ-VAE tokenizer, indicating a tokenizer-imposed bottleneck rather than a lack of decoder convergence. Results for secondary structure appear in Appendix; Table 27.

Table 8: 3D structure prediction on continuous automated model evaluation (CAMEO 2024) (Jan 2024 to Jan 2025) (Haas et al., 2018). Inference time of all methods is reported on identical A100 hardware for a representative 384-residue protein sequence. † Due to computational cost, TM-score for *AF2* methods is reported from the *ESM2* publication, using the CAMEO benchmark from April 2022 and June 2022.

| Method | TM-score | A100 Wall-clock (384-aa) | Speed-up |
|---|---|---|---|
| Prot2Token-D | 0.54 | 1–2 s | ≈1000× |
| ESMFold (ESM2-3B) (ESM Team, 2024) | 0.79 | 14.2 s | 77× |
| AF2 (w/o MSA) (Varadi et al., 2022)† | 0.41 | 20–30 s | 54× |
| AF2 (w/ MSA) (Varadi et al., 2022)† | 0.88 | 18–25 min | 1× |

### 3.5 OTHER TYPES

Building on the model's ability to predict binding sites (Section 3.3), we extended our approach to include protein-kinase phosphorylation site prediction, a task with significant real-world applications. For this, we selected protein-kinase sequence pairs along with their corresponding phosphorylation sites and jointly trained them alongside 20 self-supervised tasks. The fine-tuning phase started from the latest checkpoint obtained during the self-supervised pre-training stage. In this task, the self-supervised tasks were reduced to a total of 20,000 samples. Substrate sequences longer than 1,280 amino acids were excluded during training and evaluation. Additionally, the total sequence

length, combining substrate and kinase sequences, was capped at 2,048 tokens, with kinase sequences truncated as necessary to fit within this limit. The batch size was set to process 98,304 tokens per iteration. We enabled fine-tuning of the last eight blocks of the protein encoder.

Table 9, compares our results with two phosphorylation prediction tools, GPS 6.0 (Chen et al., 2023) and KinasePhos3 (Ma et al., 2023). Predictions with scores above 0.7 were classified as true positives. For GPS 6.0, we generated results by selecting each kinase group individually on its platform. Since the training split of GPS 6.0 is not publicly available, there is a risk of data contamination between our validation set and GPS 6.0's training data. This could result in artificially high-performance estimates for GPS 6.0, potentially reflecting memorization rather than true generalization.

Table 9: Comparative $F_1$ results of our method against leading tools (KinasePhos3 and GPS 6.0) across the validation, GPS test, and rare groups test sets.

| Method | Validation Set ($F_1$) | GPS Test Set ($F_1$) | Rare Groups Test Set ($F_1$) |
|---|---|---|---|
| KinasePhos3 (Ma et al., 2023) | 0.0747 | 0.0421 | 0.3178 |
| GPS 6.0 (Chen et al., 2023) | 0.3076 | 0.2398 | 0.4000 |
| Prot2Token-C | **0.4966** | **0.4059** | **0.4242** |

## 4 DISCUSSION

This work introduces *Prot2Token*, a unified tokenization framework that reimagines protein prediction tasks as a next token prediction. By developing a versatile tokenization strategy, we demonstrate that a single autoregressive decoder can effectively map the latent representations of pre-trained PLMs to a diverse array of biological outputs—ranging from residue-level annotations and scalar properties to complex 3D structural coordinates. This approach has the potential to represent a paradigm shift from utilizing specialized, task-specific heads to employing a general-purpose sequence generation mechanism, thereby enabling multi-task learning across completely different protein tasks that seemed impossible before.

### 4.1 KEY INSIGHTS AND EMPIRICAL OBSERVATIONS

Throughout the development and evaluation of *Prot2Token*, we observed several distinct behaviors that highlight both the capabilities and the idiosyncrasies of modeling proteins via next-token prediction.

**Unified Tokenization.** A central outcome of this study is the validation of our universal tokenization protocol. Our primary objective was not to train a single, monolithic model covering all possible downstream tasks but rather to demonstrate the feasibility of mapping the vast landscape of protein prediction problems into a cohesive generative framework. As illustrated in Figure 3, we established that virtually all protein tasks can be categorized into five structural categories: Classification, Regression, Binding Site, Sequence-to-Sequence, and Other (complex composite) tasks. By selecting and evaluating at least one representative candidate from each category, we confirmed that this unified next-token prediction format yields robust performance comparable to, and often exceeding, specialized or baseline models. This provides a scalable blueprint for bringing diverse biological tasks into the generative interface without requiring bespoke architectures for each domain.

**Hierarchical Regression.** A distinct advantage emerged from our single-digit tokenization strategy for regression tasks, which benefited most significantly from the unified framework. Unlike standard predictors that output a continuous value in a single "shot," our autoregressive approach effectively performs a coarse-to-fine prediction. By generating values digit-by-digit, the model first establishes the order of magnitude before iteratively refining the precision. This multi-step process allows for dynamic internal adjustments as the prediction becomes more granular. Consequently, *Prot2Token* outperformed the strong *ESM* + linear probe baseline—utilizing the same encoder—in nearly all regression benchmarks. As evidenced by Tables 3, 4, 6, 7, and Appendix Tables 21, 22, our method significantly surpasses the best baseline in all tasks except protein thermostability prediction (Table 23), validating that discretizing continuous spaces into hierarchical token sequences is a highly effective modeling strategy.

**Multi-Task Learning.** While investigating the full spectrum of cross-task learning (including negative transfer) was constrained by the computational cost of balancing highly diverse data distributions, we utilized protein-ligand binding site prediction as a controlled environment to study these dynamics. By treating the prediction of binding sites for 41 distinct ligands as separate tasks (each defined by a

unique prompt) we observed that aggregating tasks yielded clear synergistic benefits, particularly for data-scarce targets. Crucially, our deep investigation into the learned task token embeddings (Appendix A.5) revealed that the model encoded physicochemical correlations, grouping ligands by properties such as molecular weight and charge without explicit supervision. This semantic structure in the task space facilitated knowledge transfer, allowing the model to leverage latent patterns from over-represented ligands to significantly boost prediction accuracy for chemically similar, underrepresented ligands (Table 34). Beyond the ligand domain, we similarly observed beneficial correlations when aggregating distinct prediction types to form auxiliary tasks for a specific target, as demonstrated by the performance gains reported in Tables 2 and 7.

**Simplification and Efficiency.** A defining capability of *Prot2Token* is its ability to bring disparate and structurally complex tasks, such as kinase-specific phosphorylation (requiring multi-sequence context) and 3D structure prediction (requiring geometric reasoning), into the same simplified architectural flow. This unification renders traditionally complex prediction pipelines computationally straightforward. In the case of 3D structure prediction, this simplicity translates to an inference speedup of approximately $\sim 1000\times$ compared to AlphaFold2 (Table 8). This efficiency arises because every amino acid contributes to only one fixed computational step in the decoder's generation process, avoiding the expensive bi-directional recycling iterations of specialized models. Notably, when compared against the single-sequence (no-MSA) version of AlphaFold2, *Prot2Token* surpasses the baseline in both prediction quality and inference speed, demonstrating that general-purpose autoregressive decoders are a viable, high-throughput alternative for structure modeling.

## 4.2 LIMITATIONS

The quality and distribution of labels vary significantly across protein prediction tasks. While some datasets are uniform, others suffer from extreme imbalance; for instance, the Fold classification dataset contains classes with single samples (Appendix A.4.1), and binding site indices follow a severe long-tail distribution (Figure 9 and Figure 10). Our analysis suggests this sensitivity is a data limitation rather than an architectural flaw, as *Prot2Token* excels when data is abundant. We mitigated some of these irregularities via engineering interventions—such as token weighting and self-supervised pre-training—rather than fundamental architectural changes. This heterogeneity necessitated validating the tokenization protocol across task categories rather than pursuing a monolithic "all-task" training run. However, we anticipate this issue will diminish in real-world applications where datasets are typically magnitudes larger than academic benchmarks, thereby reducing the prevalence of extreme data label sparsity.

Furthermore, performance is intrinsically bounded by the foundational models used. Biases in the protein encoder (e.g., ESM2) propagate to predictions, for instance, the accuracy plateau in 3D structure prediction (TM-score $\approx 0.55$) reflects the reconstruction ceiling of the VQ-VAE tokenizer. Consequently, closing these gaps requires integrating higher-fidelity components rather than altering the predictor architecture.

## 4.3 FUTURE DIRECTIONS

Looking ahead, several research avenues promise to extend the capabilities of this framework. A primary technical objective is the development of high-fidelity, discrete tokenizers for 3D structures that can surpass the current reconstruction bottlenecks, potentially allowing the speed of *Prot2Token* to be paired with high-accuracy folding. Additionally, moving beyond deterministic greedy decoding to explore stochastic sampling strategies could unlock a richer landscape of probabilistic outputs, which is particularly valuable for modeling conformational diversity in structure prediction.

Perhaps the most compelling direction is the inversion of the current paradigm: extending *Prot2Token* from prediction to generation. The unified architecture naturally supports both workflows where the model could not only predict properties from sequence but also generate novel sequences conditioned on desired property tokens. This would allow for the seamless integration of prediction and design within a single cohesive model, potentially accelerating the *in silico* development of novel therapeutics and biomaterials.

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

# A APPENDIX

## A.1 ARCHITECTURE

The *predictor decoder* in Prot2Token is an autoregressive transformer that utilizes two information streams: (i) the fused encoder context, derived from protein (and optionally, chemical) embeddings processed and merged by the fusion block, and (ii) a sequence of decoder input tokens (Figure 4). The fusion block employs a straightforward architecture where, for instance, protein encoder outputs are first augmented with a learnable positional encoding and subsequently passed through a linear projection layer before being utilized by the decoder.

In the standard setting (Figure 4A), the decoder input begins with a special <BOS> token followed directly by the tokenized label sequence (e.g., the digits of a regression target). Each position attends only to previous tokens via causal masking, while simultaneously receiving global context through cross-attention to the fused encoder features. The training objective is the negative log-likelihood of the full label sequence, so loss is accumulated over every decoder position.

For multi-task training, we prepend a task token $T_i$ that specifies which prediction head the decoder should emulate (Figure 4B). This token is drawn from its own learnable embedding table and is passed through the same decoder stack as the label tokens, enabling the model to condition its hidden states on task identity. During optimization, we apply the token-weighted loss described in Section 2.2: the task token position is assigned weight $w_1 = 0$, effectively masking it from gradient updates, whereas the remaining positions use token-specific weights $w_t$, allowing each token to be penalized differently during training. This scheme enables the prompt to steer the generation process without being penalized for reconstruction errors.

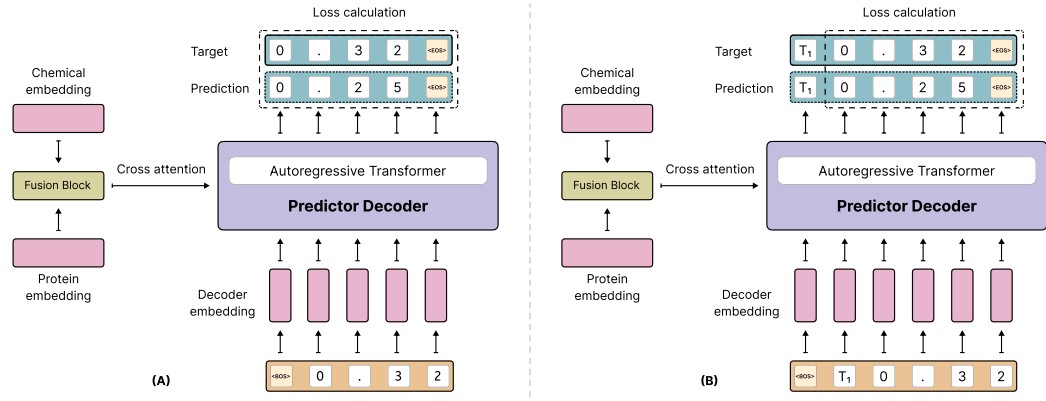

Figure 4: Task-token prompting and loss masking in the *Prot2Token* decoder. (A) Standard decoding starts with a <BOS> token and predicts label tokens, computing loss over all positions. (B) Prompted decoding inserts a task token ($T_1$) before labels; this token is zero-weighted in the loss, guiding the model without affecting training error.

Together, these mechanisms allow a single decoder to (i) handle heterogeneous output formats, (ii) switch tasks via lightweight prompt tokens, and (iii) share parameters across tasks without duplicating specialized heads.

Different configuration of *Prot2Token* is shown in Table 10.

Table 10: *Prot2Token* model configurations.

| Name | Encoder | Decoder | | | |
|------|---------|---------------------|------------------------|-------|--------|
| | | **Embedding dimension** | **Feedforward dimension** | **heads** | **Layers** |
| Prot2Token-A | ESM2-35m | 480 | 960 | 8 | 4 |
| Prot2Token-B | ESM2-650m | 640 | 1280 | 8 | 8 |
| Prot2Token-C | ESM2-650m | 640 | 2560 | 16 | 16 |
| Prot2Token-D | ESM2-650m | 1280 | 5120 | 16 | 16 |

During inference, the decoder generates tokens autoregressively, starting from the initial input (`<BOS>` or `Task token`) and predicting one token at a time. While various decoding strategies exist for autoregressive transformers—such as top-$k$ sampling, nucleus (top-$p$) sampling, and temperature-controlled sampling—we focus exclusively on *greedy decoding* in this work, where at each step the most probable token is selected. Exploring the effects of stochastic sampling methods on prediction performance is left for future investigation.

## A.2 TOKENIZATION

The *Prot2Token* framework employs distinct tokenization approaches for its input encoders and the output decoder. Input sequences, such as protein amino acid sequences or chemical *SMILES* representations, are processed using the built-in tokenizers associated with their respective pre-trained encoders. For instance, the protein encoder typically utilizes the character-level tokenizer from models like *ESM2* (which includes 33 unique tokens, encompassing standard amino acids and special characters). Similarly, if a chemical encoder is used (e.g., for protein-ligand tasks), it would employ its specific tokenizer, such as the unigram tokenizer from *BARTSmiles* (Chilingaryan et al., 2022)(with 1025 unique tokens, including special characters).

The core innovation of *Prot2Token* lies in its unified tokenization strategy developed for the output labels predicted by the autoregressive decoder. This strategy is crucial as it converts diverse biological prediction targets into sequences of discrete tokens. This conversion enables the decoder to handle a wide array of tasks through a consistent next-token prediction mechanism. All tokenized output sequences are standardized to begin with a `<BOS>` token and end with an `<EOS>` (end-of-sequence) token. These special tokens clearly define the boundaries of the output sequence for the decoder.

The specific methods for tokenizing different types of labels are categorized by the nature of the prediction task (Figure 3):

### A.2.1 CLASSIFICATION

Classification tasks involve assigning one or more categorical labels to a protein or a pair of proteins. This category includes multi-class, multi-label, and hierarchical classification.

**Multi-class classification.** In multi-class each input (single protein sequence or multiple sequences like a protein pair) is assigned exactly one label from a predefined set of mutually exclusive classes, and each possible class label is mapped to a unique, discrete token. Examples include predicting protein fold class, subcellular localization, or enzyme reaction (ER) categorization. For tasks involving interactions, such as predicting if two proteins interact (a binary classification based on a protein pair input), the output is also a single token (e.g., `Interacted`). In general, the target output sequence for the decoder is a single token representing the correct class.

**Multi-label classification.** Multi-label is employed when a single protein (or input entity) can be associated with multiple labels simultaneously from non-mutually exclusive classes. This is common in tasks such as predicting gene ontology (GO) terms (e.g., `GO:0005737` for cytoplasm, `GO:0005829` for cytosol) or certain subcellular localization tasks (e.g., DeepLoc 2.0 (Thumuluri et al., 2022)) where a protein might reside in multiple compartments. Each relevant label is converted into its unique token, and these tokens are concatenated into a single target sequence, typically sorted alphabetically to ensure consistency (e.g., `GO:0005737, GO:0005829`).

**Hierarchical classification.** In tasks such as enzyme commission (EC) and ER predictions, proteins are categorized hierarchically. For EC, each enzyme is assigned a series of numbers representing its

specific catalytic activity. If the goal is to do hierarchical classification, it necessitates a specialized tokenization approach. As an example, the EC classification system is divided into four levels: the first level indicates the main enzyme class, the second level specifies the subclass, the third level defines the sub-subclass, and the fourth level denotes the serial number of the enzyme in its sub-subclass. We tokenize each EC number associated with an enzyme into a hierarchical sequence of tokens. For example, an enzyme with EC numbers `1.1.1.1` and `2.2.2.2` is tokenized as `{ec_1, 1, 1, 1, ec_2, 2, 2, 2}`, with each part of the EC number being represented as an individual token. This approach allows the model to capture the hierarchical nature of enzyme classifications effectively, ensuring that the different levels of EC labels are properly represented and learned. In addition to this hierarchical tokenization, we could employ a second approach where each complete EC number is treated as a unique and distinct code similar to GO datasets. For example, an enzyme with EC numbers `1.1.1.1` and `3.4.24.4` could be tokenized as `{ec_1.1.1.1, ec_3.4.24.4}`, with each token acting as a representative for an entire EC number. This method is also applicable to the ER dataset. This alternative tokenization could yield different results depending on the task. In our early experiments, we found that converting ER labels into a hierarchical format reduced performance compared to using a multi-label classification format, while the opposite was true for the EC task. However, we did not investigate this thoroughly in our work.

### A.2.2 REGRESSION

Regression tokenization is employed for tasks requiring the prediction of continuous numerical values, represented as either floating-point or integer numbers, derived from single protein sequences, multiple sequences, or protein-ligand pairs. Illustrative examples include the prediction of protein stability ($\Delta T_m$), fluorescence intensity (single sequence input), protein-protein structure similarity scores (multi-sequence input), and protein-ligand affinity (protein sequence and ligand *SMILES* string as input). Two primary strategies exist for tokenizing such numerical labels. The first, binning, involves discretizing the range of continuous values into a predefined number of fixed-size bins. For instance, if target scores range from 0.0 to 10.0, this range could be divided into 1.0-sized bins, yielding 11 distinct token categories. However, this method can suffer from limitations, particularly when data is unevenly distributed, as some bins may contain very few or no samples, leading to imbalanced data representation and potential biases during model training. To circumvent these issues, *Prot2Token* adopts a second approach: a digit-by-digit encoding strategy. In this method, each numerical value is transformed into a sequence of its constituent characters, including the sign, digits, and decimal point. This technique offers a more granular and inherently balanced representation of numerical values, promoting a more uniform distribution of data for the model. For example, a property value of $-0.65$ is tokenized into the sequence `{minus, 0, ., 6, 5}`. Similarly, a value of `123.45` would become `{1, 2, 3, ., 4, 5}`. During the training phase, a consistent numerical precision, typically four decimal places, is maintained for all regression labels prior to tokenization. Furthermore, if target values undergo normalization (e.g., to the `[0, 1]` range), the token sequences predicted by the decoder are first reconverted to numerical form and subsequently de-normalized to their original scale for evaluation.

We investigated the impact of token ordering on regression performance by reversing the target sequence to a right-to-left format (least significant digit first). We observed a measurable degradation in performance compared to the standard left-to-right approach. We attribute this to the loss of the coarse-to-fine inductive bias inherent in left-to-right generation, where the model first predicts the most significant digits (establishing magnitude) before refining the value with lower-order precision; reversing this order forces the model to predict fine-grained details without an established context for the overall scalar value, leading to overfitting and reduced accuracy.

### A.2.3 SEQUENCE-TO-SEQUENCE

This tokenization is applied when the output is a sequence of labels corresponding residue-by-residue to the input protein sequence, meaning the output token sequence length mirrors the input protein length. Examples include Secondary Structure (SS) prediction, where each amino acid is classified into states like $\alpha$-helix (`H`), $\beta$-strand (`E`), or coil (`C`), forming a target sequence like `{H, H, C, ...}`. Another application is 3D structure prediction using structural alphabets. For instance, the encoder part of a pre-trained VQVAE) model (Gaujac et al., 2024) converts 3D coordinates into a sequence of discrete `3D_number` tokens (e.g., 4096), where each amino acid corresponds to one `3D_i` token encoding 3D structural information.

### A.2.4 BINDING SITE

Binding site prediction involves identifying specific residue indices involved in molecular interactions, such as with ligands, ions, or other proteins. For protein-ligand or protein-ion binding tasks, the binding residues are represented directly by their sorted 1-based indices; for example, if residues at positions 2, 3, 5, and 9 are involved in binding, the target sequence is simply {2, 3, 5, 9}. Self-supervised learning tasks proposed in this work also utilizes index-based tokenization. For example, to predict the positions of all Serine (S) residues in a sequence MSGLSNYT (Serines at positions 2 and 5), the target sequence would be {2, 5}. Twenty such tasks can be created, one for each standard amino acid, helping the decoder learn sequence-position relationships.

### A.2.5 OTHER TYPES

This category includes tasks like PTMs prediction and tasks that combine different output types.

**PTMs.** This involves identifying potential modification sites and the actual modified sites. For a single protein sequence input, the target sequence typically lists the 1-based indices of all potential PTM sites (e.g., S, T, Y for phosphorylation), followed by a special separator token (<SEP>), and then the 1-based indices of experimentally confirmed positive sites, all sorted numerically. For example, for a sequence ASSKYKAMTV, phosphorylation prediction might yield {2, 3, 5, 9, <SEP>, 3, 9}. In multi-sequence PTM tasks, such as substrate-kinase phosphorylation prediction, the input consists of both substrate and kinase sequences. The output tokenization still focuses on the substrate, listing potential and confirmed phosphorylation sites on the substrate sequence based on the interaction context provided by the kinase.

**Combination.** Tasks like TargetP 2.0 (Armenteros et al., 2019) combine classification and regression. For instance, a label might be represented as {sp, 96}, where sp is a localization class token (Signal Peptide) and 96 is a binding site representing the cleavage site position. This is tokenized by concatenating these two types.

### A.3 DATASET

To assess *Prot2Token* across a representative spectrum of protein–biology problems, we assembled datasets from several public repositories and task-specific benchmarks. The statistics of each task are shown in Table 11.

Table 11: Dataset Statistics Overview. This table presents the details of the datasets utilized in this study. †: Randomly 300k of samples are used for the training in each fold.

| Dataset | Train | Validation | Test | Task Type |
|---|---|---|---|---|
| Enzyme commission (Omelchenko et al., 2010) | 15,550 | 1,720 | 1,919 | Classification |
| Gene ontology (Consortium, 2008) | 29,898 | 3,322 | 3,415 | Classification |
| Fold classification - Fold (Hou et al., 2018) | 12,312 | 736 | 718 | Classification |
| Enzyme reaction (Webb et al., 1992) | 29,215 | 2,562 | 5,651 | Classification |
| Human PPI (Xu et al., 2022) | 35,669 | 315 | 237 | Classification |
| DeepLoc 2.0 (Thumuluri et al., 2022) | 22,841 | 5,462 | 1,717 | Classification |
| Kinase group classification (Chen et al., 2023) | 5,382 | 969 | - | Classification |
| Mutation stability (Notin et al., 2023) | ≈1.92 million† | ≈480,000 (5-fold) | - | Regression |
| Structure similarity(Kucera et al., 2023) | 300,700 | 4,560 | 4,851 | Regression |
| Protein-ligand affinity (Xu et al., 2022) | 16,436 | 937 | 285 | Regression |
| Protein-protein binding affinity (Liu et al., 2024) | 765 | 180 | 270 | Regression |
| Stability (Xu et al., 2022) | 53,571 | 2,512 | 12,851 | Regression |
| Fluorescence (Xu et al., 2022) | 21,446 | 5,362 | 27,271 | Regression |
| Thermostability (Chen & Gong, 2022) | 131,260 | 14,584 | 36,461 | Regression |
| Protein-protein binding site (Bushuiev et al.) | 759,282 | 2,918 | 5,499 | Binding site |
| Protein-ligand binding site (Bushuiev et al.) | 16,796 | 2,644 | 5,153 | Binding site |
| Structure prediction (Varadi et al., 2022) | 10,876,251 | 5,000 | 5,000 | Sequence to sequence |
| Secondary structure (Xu et al., 2022) | 8,678 | 2,170 | 513 | Sequence to sequence |
| Target-P 2.0 (Armenteros et al., 2019) | 10,400 | 2,605 | - | Other (classification, regression) |
| PTMs | Table 12 | Table 12 | Table 12 | Other (PTM) |
| Kinase phosphorylation (Chen et al., 2023) | 5,382 | 969 | 146 | Other (PTM) |

### A.3.1 PEER BENCHMARK

The PEER benchmark (Xu et al., 2022) provides a unified evaluation suite for protein sequence understanding, integrating datasets for protein function, subcellular localisation, secondary structure,

protein–protein interaction (PPI), and protein–ligand affinity prediction. Each task is delivered with homology-aware train/validation/test splits and pre-defined evaluation metrics, enabling direct comparison between conventional feature-engineering pipelines, sequence-embedding models, and large-scale protein language models. From PEER we adopt five datasets that align with our experimental focus: (i) human PPI pairs for binary interaction prediction, (ii) secondary-structure assignments for residue-level sequence-to-sequence labelling, (iii) fluorescence intensities for single-sequence regression, (iv) stability ($\Delta T_m$) measurements for mutation-effect regression, and (v) protein–ligand affinity (PLA) scores for sequence–SMILES binding prediction.

### A.3.2 DEEPLOC 2

For subcellular localization we adopted the DeepLoc 2.0 dataset (Thumuluri et al., 2022), which assigns up to ten compartment labels per eukaryotic protein: *Cytoplasm*, *Nucleus*, *Extracellular*, *Cell membrane*, *Mitochondrion*, *Plastid*, *Endoplasmic reticulum*, *Lysosome/Vacuole*, *Golgi apparatus*, and *Peroxisome*. DeepLoc 2.0 provides a five-fold homology partition with a maximum 30 % pairwise sequence identity between folds. In our experiments the first four folds are merged for training, while the fifth fold serves as the validation set. Evaluation is performed on the independent Human Protein Atlas (HPA) test set released with DeepLoc 2.0, which contains experimentally verified localizations for six compartments (*Cytoplasm*, *Nucleus*, *Cell membrane*, *Mitochondrion*, *Endoplasmic reticulum*, and *Golgi apparatus*). Final performance is reported on this HPA test set.

### A.3.3 PTMS

In this section, we describe the process of collecting PTM data. While numerous databases and publications provide PTM data, most only offer sequence fragments, typically 21 amino acids long, with the PTM located at the center position. The largest database with PTM annotations is UniProt (Consortium, 2019), which contains over 200 million protein sequences and provides annotations for more than 200 PTM types and their respective positions for some sequences. We downloaded full-length protein sequences and PTM annotations from UniProt, focusing on annotations in the *Modified Residue*, *Lipidation*, *Glycosylation*, and *Cross-link* sections and performed an advanced search in these sections using a wildcard (*) to retrieve all values. This resulted in 106,195 protein sequences from the Reviewed (Swiss-Prot) (Boeckmann et al., 2003) dataset and 4,173,205 sequences from the Unreviewed (TrEMBL) dataset. To ensure data quality, we exclusively used the protein sequences from the Reviewed (Swiss-Prot) dataset.

We downloaded the 106,195 protein sequences as JSON files for further processing, only sequences with lengths of 1,022 amino acids or fewer were retained. Next, CD-HIT (Li & Godzik, 2006) was applied to cluster the sequences based on a similarity threshold of 40% (`c=0.4`), grouping sequences with similarity above 40% into the same cluster. Subsequently, we split the data into training and testing sets in a 4:1 ratio, ensuring that sequences within the same cluster were assigned to the same dataset. Given the distribution of PTM types, we focused on six types for this study: Phosphorylation (S), Methylation (R), N-glycosylation (N), O-glycosylation (T), Acetylation (K), and Ubiquitylation (K).

Table 12 shows the statistics of the PTM datasets.

Table 12: Statistics of PTM datasets.

| PTM type | Annotation in *Uniprot* | Amino acid | Number of sequences | Number of positions |
|---|---|---|---|---|
| Ubiquitylation | Glycyl lysine isopeptide (Lys-Gly) (interchain with G-Cter in ubiquitin) | K | 2,370 | 5,029 |
| Phosphorylation | Phosphoserine | S | 34,260 | 121,398 |
| Acetylation | N6-acetyllysine | K | 9,115 | 23,615 |
| Methylation | Omega-N-methylarginine | R | 1,813 | 3,279 |
| N-linked Glycosylation | N-linked (GlcNAc...) asparagine | N | 30,310 | 11,576 |
| O-linked Glycosylation | O-linked (GalNAc...) threonine | T | 568 | 2,723 |
| Succinylation | N6-succinyllysine | K | 2,392 | 7,446 |

### A.3.4 KINASE-SPECIFIC PHOSPHORYLATION SITES

The dataset was gathered from GPS 6.0 (Chen et al., 2023) and contains 24,160 phosphorylation sites. We mapped IDs from the UniProt database (Consortium, 2019) and obtained 13,374 sequences with kinase information. To retrieve kinase sequences, we used Kinase.com and the UniProt database. To reduce sequence similarity, we applied CD-HIT (Li & Godzik, 2006) with a 70% similarity threshold to group similar protein substrate sequences. We kept representatives from each cluster and

selected positive substrate-kinase pairs using two criteria: (1) cross-cluster selection, where pairs from different clusters were kept to increase diversity, and (2) within-cluster selection, where only one unique kinase pair per cluster was retained to avoid repetition. The final dataset includes kinase sequences, kinase information (group/family/kinase), substrate UniProt IDs, substrate sequences, and phosphorylation sites. It contains 386 kinase types across 12 groups.

The dataset was randomly split into training (5,382 unique substrates) and validation (969 unique substrates) sets. To ensure rigorous evaluation, we defined three distinct test sets, carefully designed to prevent any data contamination between the test, training, and validation sets:

**Rare-Group Test Set.** This set includes 14 samples from two rare kinase groups, *'RGC'* and *'PKL'*, which have a limited number of available samples. These groups were completely excluded from the training set to assess the model's ability to generalize to underrepresented kinase groups. This dataset is specifically used for evaluating on phosphorylation site prediction.

**GPS-Test Set.** To have a direct comparison with existing methods such as GPS 6.0, we adopted the test set used in the GPS study. This dataset contains 146 samples of substrate-kinase pairs, including phosphorylation site and kinase group annotations. All samples belong to the *'CMGC'* kinase group. Table 13 presents the number of samples in each set, while Table 14 details the distribution of samples across kinase groups in each dataset.

Table 13: Dataset statistics, including the number of samples, phosphorylation sites (p-sites), and kinase groups for the training, validation, GPS test, and rare group test sets, along with overall dataset totals.

| Dataset | Number of samples | Number of p-sites | Number of groups |
|---|---|---|---|
| All samples | 6,511 | 13,374 | 12 |
| Training set | 5,382 | 10,621 | 10 |
| Validation set | 969 | 2,455 | 9 |
| GPS-test | 146 | 278 | 1 |
| Rare-Group | 14 | 25 | 2 |

Table 14: Distribution of samples across kinase groups for the training, validation, GPS test, and rare group test sets.

| Group | Training set | Validation set | GPS-test | Rare-Group |
|---|---|---|---|---|
| AGC | 1,446 | 231 | - | - |
| Atypical | 270 | 58 | - | - |
| CAMK | 653 | 96 | - | - |
| CK1 | 100 | 27 | - | - |
| CMGC | 1,466 | 264 | 146 | - |
| Other | 491 | 99 | - | - |
| STE | 211 | 34 | - | - |
| TK | 677 | 149 | - | - |
| TKL | 68 | 11 | - | - |
| RGC | - | - | - | 2 |
| PKL | - | - | - | 12 |

### A.3.5   PROTEIN MUTATION STABILITY

In this study, we used the supervised Deep Mutational Scanning (DMS) cross-validation subset of the ProteinGym (Notin et al., 2023) benchmark, a large-scale and standardized resource for evaluating protein fitness prediction models. The supervised DMS dataset comprises over 250 high-throughput assays, covering more than 2.4 million amino acid substitutions across 217 proteins, and approximately 300,000 indel mutations across 66 proteins. Each assay provides experimentally measured phenotypic effects for a wide range of mutations, reflecting properties such as thermostability, binding affinity, aggregation, and viral replication. We followed the five-fold cross-validation indices defined by ProteinGym, conducting five independent training runs, each on a 300,000-sample subset of the full dataset due to computational constraints. ProteinGym categorizes benchmarks by mutation type (substitutions vs. indels) and ground-truth source (DMS assays vs. clinical annotations); in this work, we utilized only substitution dataset within the supervised regime.

### A.3.6 Protein melting temperature

We leveraged the HotProtein (Chen & Gong, 2022) sequence-only benchmark to predict protein melting temperatures from primary sequence alone. HotProtein comprises 182,000 amino acid sequences of 230 organisms, each labeled with the optimal growth temperature of its source organism (-20°C to 120°C) as a lower bound proxy of the true melting temperature of the protein. For evaluation, the ProCeSa (Zhou et al., 2025) paper defined 10-fold cross-validation splits on various subsets of the dataset, such as HP-S2C2 (binary: hot vs. cold), HP-S2C5 (five-class), and HP-S (full dataset). In our study, we used only the first fold of the provided splits, further dividing the training portion into training and validation sets.

### A.3.7 3D structure prediction

For training our model on sequence-to-structure prediction, we constructed a large-scale dataset from UniRef50 (Consortium, 2019) , a redundancy-reduced cluster of protein sequences derived from UniProt. This provided approximately 67 million unique sequences. We mapped these sequences to their predicted structures using the UniProt *AF2* Structural Database (Varadi et al., 2022), yielding 40 million PDB files. To ensure high structural confidence, we filtered out structures with mean pLDDT scores below 0.85, resulting in about 11 million high-confidence entries. From this filtered pool, we randomly selected 5,000 PDBs each for validation and test sets, ensuring all selected structures had average pLDDT scores above 0.90. The remaining structures were used for training. All 3D structures were converted into discrete token sequences using a pre-trained VQ-VAE model (Gaujac et al., 2024), enabling their use as target labels for autoregressive sequence-to-structure modeling.

The continuous automated model evaluation (CAMEO) (Haas et al., 2018) platform offers continuous, automated benchmarking of protein structure prediction methods by evaluating their performance on newly released target sequences each week, providing a real-time complement to the biennial CASP experiment. In this study, we used CAMEO targets released between January 2024 and January 2025, comprising 668 protein sequences. After filtering for sequences between 50 and 512 amino acids in length, the final dataset contained 576 sequences.

### A.3.8 Protein-protein affinity

We used data from PPB-Affinity (Liu et al., 2024), the largest publicly available dataset for protein-protein binding (PPB) affinity. PPB-Affinity provides key information, including crystal structures of protein-protein complexes, PPB affinity values, receptor protein chains, and ligand protein chains. Since PPB-Affinity does not include protein sequences, we retrieved them from the RCSB Protein Data Bank (PDB) (Berman et al., 2000) based on the protein names provided in PPB-Affinity. To construct a relevant dataset for our model, we applied the following filtering steps:

1. **Chain Filtering** – We removed samples containing more than two chains, retaining only those with a single receptor chain and a single ligand chain.

2. **Mutation Removal** – Samples containing mutated sequences were excluded.

3. **Affinity Label Processing** – For identical protein complexes with multiple PPB affinity values, we averaged the KD (M) values to obtain a single affinity label.

4. **Data Splitting** – The final dataset was split into training (50%), validation (20%), and testing (30%) sets, resulting in 765, 180, and 270 samples, respectively.

The $(KDK_D)$ values, representing dissociation constants, were preprocessed to ensure numerical stability and improve model performance. First, a log10 transformation was applied to address the wide dynamic range and skewed distribution of KD values, using the formula: $KD_{\log} = \log_{10}(KD + \epsilon)$, where $\epsilon = 10^{-16}$ prevents undefined values for extremely small inputs. The log-transformed values were then normalized to a range between 0 and 1 using Min-Max scaling based on the training dataset's minimum and maximum $KDlog_{\log}$ values. Importantly, during model metric calculation and evaluation, both the log-transformation and normalization effects were reversed, ensuring that the calculated metrics accurately reflect the original KD scale. This preprocessing pipeline provided a consistent and interpretable representation of KD values for both model training and evaluation.

### A.3.9 Gene Ontology

The GO knowledge-base provides curated associations between protein sequences and hierarchically organized terms spanning three sub-ontologies: Molecular Function, Biological Process and Cellular Component (Consortium, 2008). We downloaded the most recent GOA-UniProt annotation file, removed electronically inferred codes (IEA) and retained only leaf-level terms, yielding a multi-label dataset in which each protein can carry dozens of GO terms. Following the convention in (Consortium, 2008), proteins were clustered at 30 % global sequence identity with MMseqs2; clusters were then split 80 / 10 / 10 into training, validation and test partitions to avoid homologous leakage. Labels are tokenised as individual GO identifiers in alphabetical order, in accordance with the scheme in Section 2.2.

### A.3.10 Enzyme reaction

The ER corpus collates detailed reaction schemata and catalytic-site annotations for enzymes, originally introduced by Webb (Webb et al., 1992). We used the reaction–protein mappings distributed via EzCatDB (Nagano, 2005), which capture bond-level changes and catalytic residue motifs. Each protein may participate in multiple reactions, making ER a multi-label classification task. Sequences were clustered at 40 % identity and split into 70 % training, 15 % validation and 15 % test sets. Reaction identifiers were tokenised as discrete labels; hierarchical relations (substrate → product) are ignored in this work.

### A.3.11 Enzyme commission

The EC hierarchy assigns a four-level numerical code to every known enzymatic function (Omelchenko et al., 2010). We retrieved the full set of Swiss-Prot entries with experimentally verified EC numbers from the UniProt "enzyme.dat" archive (Omelchenko et al., 2010). Proteins were redundancy-reduced at 40 % identity and stratified into train/val/test splits by superfamily. Tokenisation follows the hierarchical scheme in Section 2.2: each digit of the EC code is emitted as an independent token (e.g. `1,1,1,1`). This framing yields a four-step sequence-to-sequence prediction task.

### A.3.12 Fold classification

For remote-homology evaluation we use the dataset released with DeepSF (Hou et al., 2018), which maps protein sequences onto 1,195 folds derived from CATH (Wang et al., 2025a) and SCOP. The authors provide non-redundant splits with a maximum 40 % sequence identity between training (12,312 proteins), validation (736 proteins) and test (718 proteins) sets(Hou et al., 2018). Each fold ID is tokenised as a single class token, rendering the task a large-scale multi-class classification benchmark.

### A.3.13 TargetP 2.0 localization

TargetP 2.0 offers a homology-partitioned dataset for predicting N- or C-terminal targeting peptides and corresponding subcellular localizations (Armenteros et al., 2019). We downloaded the FASTA sequences and label CSVs from the official service repository. After filtering fragments and sequences shorter than 50 residues, the data comprise ten localization classes (*Chloroplast*, *Mitochondrion*, *Secretory pathway*, etc.), with an external HPA test set for human proteins(Armenteros et al., 2019). We adhere to the original nested cross-validation splits for training and use the HPA subset exclusively for final evaluation, casting the task as multi-class prediction with one token per localization label.

### A.3.14 Protein-ligand binding site

BioLip2 (Zhang et al., 2024) is one of the most comprehensive databases for ligand-protein interactions, primarily derived from the PDB database. Each entry in BioLip2 includes detailed annotations on ligand-binding residues, ligand-binding affinities, catalytic site residues, EC numbers and GO terms. The database is also cross-linked with external resources, including RCSB PDB, UniProt, and PubMed. To obtain protein sequences, we used receptor sequences clustered at a 90% identity cutoff. For annotations, we retrieved data for each ligand-protein interaction site. To increase the complexity of binding site prediction and enhance model robustness, we further clustered the data at

a 40% identity cutoff. This additional clustering step helps prevent data leakage between training, evaluation, and testing datasets. We first removed DNA and RNA sequences and excluded any sequences with fewer than 50 residues. Next, we generated FASTA files containing residues and annotations for all 5,717 ligands. We then applied a threshold cutoff, selecting ligands that bind with over 100 sequences, resulting in 41 ligands. We aimed to balance selecting the most significant ligands based on a literature review while ensuring a sufficient number of samples for training and testing the model. We used CD-HIT to cluster the data with a 40% identity cutoff before splitting the data into training, evaluation, and testing datasets. Because of the limited number of samples and to ensure sufficient data for testing, we used two splitting ratios: 70%, 10%, and 20% for training, evaluation, and testing, respectively, for the first 30 ligands in Table 15, and also, 50%, 20%, and 30% for training, evaluation, and testing, respectively, for the remaining ligands.

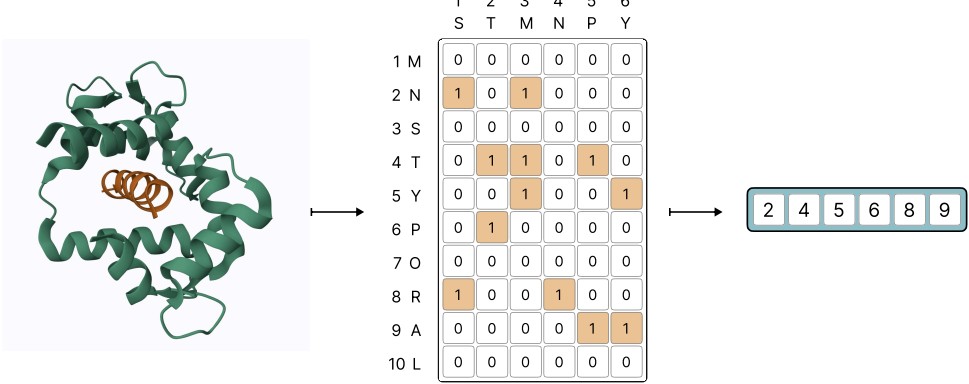

Figure 5: Tokenisation workflow for protein–protein binding sites. A distance cut-off is applied to a residue–residue distance matrix derived from the PDB complex to flag contacting residues. Rows with at least one contact are collapsed into a sorted list of residue indices, which becomes the target token sequence.

### A.3.15 PROTEIN-PROTEIN BINDING SITE

To construct a dataset for protein binding site prediction, we used the PPIRef (Bushuiev et al.) pair dataset, which specifies interacting protein chain pairs based on a contact threshold. To ensure high-quality and complete data, we retrieved all corresponding PDB entries from the PDB databas and extracted the relevant chains based on PPIRef annotations. For each protein complex, we extracted the amino acid sequences and computed residue-level binding sites by analyzing spatial proximity. Specifically, we calculated the centroid of each residue by averaging the atomic coordinates (excluding hydrogens), then computed a pairwise distance matrix between all centroids from the two chains. Residues were labeled as binding site residues if any cross-chain centroid distance fell below a 6Å threshold (Figure 5). To augment the dataset, we alternated which chain was considered the "target" and which was the "binder" in each complex. The resulting dataset includes the fields: PDB ID, Target Chain, Binder Chain, Target Sequence, Binder Sequence, Target Binding Sites, and Binder Binding Sites. For training and evaluation, we performed a randomized split grouped by PDB IDs, ensuring that each PDB complex appears in only one of the train, validation, or test sets to avoid data leakage.

### A.3.16 PROTEIN-PROTEIN STRUCTURE SIMILARITY

ProteinShake (Kucera et al., 2023) is a Python toolkit developed to streamline dataset construction and benchmarking in protein structure-based deep learning. It supports both custom and pre-processed datasets sourced from the PDB database and AFDB, and associates each dataset with well-defined prediction tasks and evaluation metrics. The framework includes standardized data splits based on sequence and structural similarity, enabling rigorous and reproducible comparisons across models and modalities (e.g., graphs, voxel grids, and point clouds). In this work, we adopt the protein-protein structure similarity dataset provided by ProteinShake and follow their *Structure Split* protocol,

applying a 70% similarity threshold to partition the data for evaluation. Notably, we use only the protein sequence information as input and do not leverage 3D structural features.

Table 15: Dataset statistics of all ligands.

| Ligand No | Chemical Formula | Name | BioLip Fasta Name | Num Sequences | Binding Sites |
|---|---|---|---|---|---|
| 1 | $Zn^{2+}$ | Zinc Ion | ZN.fasta | 4665 | 23310 |
| 2 | $CA^{2+}$ | Calcium Ion | CA.fasta | 3043 | 22161 |
| 3 | CLA | Chlorophyll A | CLA.fasta | 342 | 17690 |
| 4 | FAD | Flavin-Adenine Dinucleotide | FAD.fasta | 825 | 16583 |
| 5 | HEM | Heme | HEM.fasta | 845 | 13118 |
| 6 | NAD | Nicotinamide Adenine Dinucleotide | NAD.fasta | 658 | 10615 |
| 7 | ADP | Adenosine Diphosphate | ADP.fasta | 941 | 9899 |
| 8 | $MG^{2+}$ | Magnesium Ion | MG.fasta | 2951 | 9494 |
| 9 | NAP | Nicotinamide Adenine Dinucleotide Phosphate | NAP.fasta | 462 | 8108 |
| 10 | ATP | Adenosine Triphosphate | ATP.fasta | 680 | 7635 |
| 11 | HEC | Heme C | HEC.fasta | 264 | 7296 |
| 12 | SF4 | Iron/Sulfur Cluster | SF4.fasta | 509 | 5834 |
| 13 | FMN | Flavin Mononucleotide | FMN.fasta | 437 | 5789 |
| 14 | SAH | S-Adenosyl-L-Homocysteine | SAH.fasta | 392 | 4675 |
| 15 | NDP | Nucleotide Diphosphate | NDP.fasta | 243 | 4301 |
| 16 | ANP | Adenylyl-imidodiphosphate | ANP.fasta | 354 | 3861 |
| 17 | GDP | Guanosine Diphosphate | GDP.fasta | 339 | 3792 |
| 18 | GLC | Glucose | GLC.fasta | 454 | 3674 |
| 19 | PLP | Pyridoxal-5'-Phosphate | PLP.fasta | 377 | 3608 |
| 20 | $MN^{2+}$ | Manganese Ion | MN.fasta | 789 | 3315 |
| 21 | COA | Coenzyme A | COA.fasta | 259 | 2870 |
| 22 | SAM | S-Adenosylmethionine | SAM.fasta | 214 | 2540 |
| 23 | AMP | Adenosine Monophosphate | AMP.fasta | 275 | 2430 |
| 24 | BGC | Beta-D-Glucose | BGC.fasta | 331 | 2375 |
| 25 | $FE^{3+}$ | Ferric Ion | FE.fasta | 532 | 2268 |
| 26 | MAN | Mannose | MAN.fasta | 446 | 2047 |
| 27 | FES | Iron-Sulfur Cluster | FES.fasta | 272 | 1986 |
| 28 | $PO_4^{3-}$ | Phosphate Ion | PO4.fasta | 378 | 1908 |
| 29 | GTP | Guanosine Triphosphate | GTP.fasta | 150 | 1724 |
| 30 | UDP | Uridine Diphosphate | UDP.fasta | 154 | 1601 |
| 31 | $CU^{2+}$ | Copper Ion | CU.fasta | 331 | 1530 |
| 32 | GSH | Glutathione | GSH.fasta | 200 | 1516 |
| 33 | AGS | Agmatine Sulfate | AGS.fasta | 136 | 1512 |
| 34 | ACO | Aconitase | ACO.fasta | 108 | 1482 |
| 35 | GAL | Galactose | GAL.fasta | 233 | 1188 |
| 36 | $SO_4^{2-}$ | Sulfate Ion | SO4.fasta | 218 | 1177 |
| 37 | CLR | Cholesterol | CLR.fasta | 176 | 1112 |
| 38 | Y01 | Cholesterol Hemisuccinate | Y01.fasta | 106 | 991 |
| 39 | BMA | Beta-Mannose | BMA.fasta | 158 | 696 |
| 40 | $FE^{2+}$ | Ferrous Ion | FE2.fasta | 186 | 675 |
| 41 | $CO^{2+}$ | Cobalt Ion | CO.fasta | 160 | 660 |

## A.4 EXPERIMENTS

### A.4.1 CLASSIFICATION

For the Fold classification task, we maintained the *ESM* model weights as fixed and only unlocked its last six layers to be fine-tuned and connected to the decoder. Many classes in this dataset have a low number of samples, e.g., one sample for a high number of classes. That is why we saw unstable training when we did single-task training on *Prot2Token*. However, when we combined Fold classification with auxiliary tasks like ER, the training became stable (Table 29).

Table 16: Fold classification training in single-task and multi-task training on Fold-fold test set.

| Method | Aux-Tasks | Accuracy |
|---|---|---|
| Baseline (ESM2-650m) | - | 32.87 |
| Prot2Token-B | - | N/A |
| Prot2Token-B | ER | 31.47 |

Regarding the Human PPI task, we maintained the *ESM* model weights as fixed and only unlocked the last four layers of it to be fine-tuned and connected to the decoder. Note that to give the encoder two sequences at one feed for PPI, we concatenated two sequences using the <EOS> token. We observed that adding more tasks helped boost the performance of Human PPI (Table 16). However, *Prot2Token* tended to overfit on this task, indicating that the improvement from adding auxiliary tasks may be due to the regularization effect of multi-task learning. We used early stopping to avoid overfitting.

Table 17: Human PPI performance on PEER test set.

| Method | Aux-Tasks | Accuracy | Encoder |
|---|---|---|---|
| PEER (Xu et al., 2022) (fine-tuned) | - | 78.17 | ESM1-1b |
| Prot2Token-B | - | 71.3 | ESM2-650m |
| Prot2Token-B | Deeploc | 78.48 | ESM2-650m |
| Prot2Token-B | Deeploc+ER+Fold | 80.17 | ESM2-650m |

For the GO and EC tasks, we encountered a limitation in calculating the Fmax metric, which is commonly used for performance evaluation in these tasks. Instead, we used accuracy and $F_1$ score to assess our model's performance. Consequently, we were unable to directly compare our results with those of other methods that report their performance in terms of Fmax. This discrepancy highlights a significant challenge in benchmarking our approach against existing methods. The GO tasks are further divided into three categories: biological process (BP), molecular function (MF), and cellular component (CC). We jointly trained all four tasks (the three GO tasks and the EC task) together in a multi-task learning manner. Detailed performance metrics for these tasks are presented in Table 17. We maintained the *ESM* model weights as fixed and only unlocked the last four layers of it to be fine-tuned and connected it to the decoder and a linear classifier for *Prot2Token*. Note that labels in these tasks are highly imbalanced.

Table 18: Comparing GO and EC tasks with the baseline on accuracy and $F_1$ score metrics. The baseline is a linear evaluation of *ESM*. All methods are based on *ESM2-650m*.

| Method | Task | Accuracy | $F_1$ Score |
|---|---|---|---|
| Baseline | EC | 99.79 | 0.5383 |
| Baseline | GO-BP | N/A | 0.0043 |
| Baseline | GO-MF | N/A | 0.1028 |
| Baseline | GO-CC | N/A | 0.1327 |
| Prot2Token-B | EC | 99.85 | 0.6796 |
| Prot2Token-B | GO-BP | 95.88 | 0.0103 |
| Prot2Token-B | GO-MF | 97.20 | 0.0116 |
| Prot2Token-B | GO-CC | 95.35 | 0.0089 |

Next, we aimed to predict kinase groups based on substrate sequences. Specifically, we investigated how much information about the related kinase groups the model can infer solely from substrate sequences. To achieve this, we considered our processed training and validation datasets (refer to Appendix A.3), assigning multi-label classification labels by removing *Unknown*, *RGC*, *PKL*, and *UNK* samples from the training set and merging the remaining nine kinase groups associated with each substrate. The model takes a substrate sequence as input and predicts the corresponding kinase groups in alphabetical order. We allow *Prot2Token* to fine-tune the weights of the last 6 blocks of the protein encoder (*ESM2-650m*). After convergence, the decoder achieves the per-group $F_1$ scores listed in Table 19. Despite receiving no kinase information at inference time, *Prot2Token* recovers group memberships with a macro-averaged $F_1$ of 0.54, confirming that substrate context alone encodes considerable family-specific signal.

Table 19: Per-group $F_1$ scores for substrate-only kinase group classification via *Prot2Token-C*.

| Group | AGC | Atypical | CMGC | CAMK | CK1 | Other | STE | TK | TKL |
|---|---|---|---|---|---|---|---|---|---|
| $F_1$ | 0.555 | 0.493 | 0.634 | 0.420 | 0.539 | 0.605 | 0.357 | 0.674 | 0.500 |

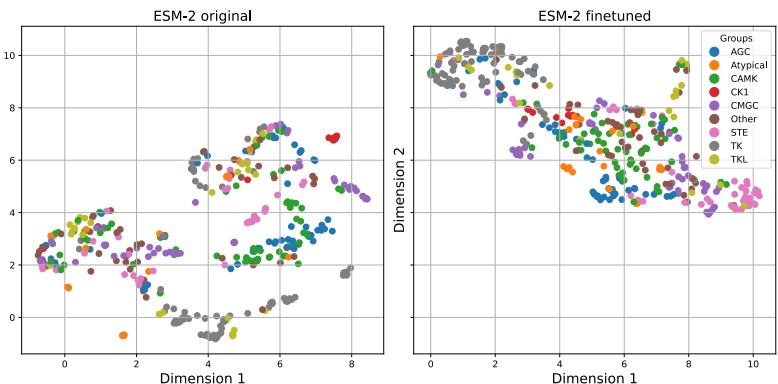

Figure 6: UMAP visualization of unique kinase sequences on the original and fine-tuned checkpoints of *ESM2-650m*.

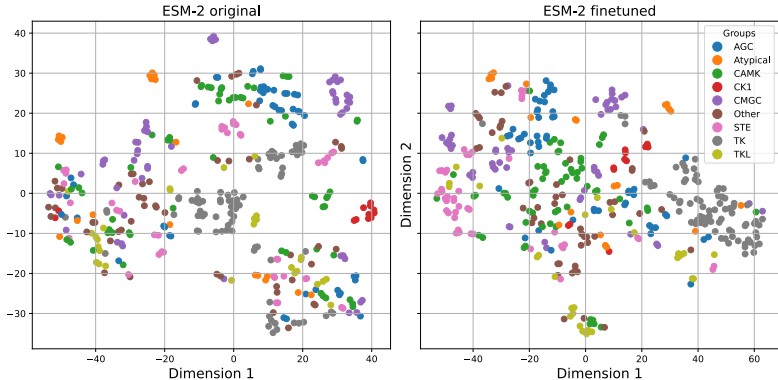

Figure 7: t-SNE visualization of unique kinase sequences on the original and fine-tuned checkpoints of *ESM2-650m*.

Table 20: Unsupervised clustering metrics for kinase embeddings. Larger values denote better separability.

| Metric | Original | Fine-tuned |
|---|---|---|
| Silhouette (cosine) | $-0.039$ | 0.091 |
| Calinski–Harabasz | 7.00 | 24.10 |

To further interpret the kinase group classification results, we analyzed the sequence embeddings of all unique kinase sequences present in the GPS 6.0 dataset before and after fine-tuning the protein encoder part of Prot2Token-C on the kinase group classification labels. Sequences from the *RGC*, *PKL*, and *Unknown* groups were excluded. Each remaining kinase sequence was passed through the pre-trained *ESM2-650m* model with a maximum input length of 2048. Token-wise embeddings were extracted, then trimmed to remove the <BOS> and <EOS> tokens, and average pooling was applied to yield a fixed-length 1280-dimensional representation per sequence, aligned with the model's hidden size.

We performed dimensionality reduction using t-SNE and UMAP to visualize these embeddings in two dimensions based on their known group assignments. Visualizations revealed that the original pretrained model exhibited weak separation among groups. However, when we repeated the same process using the fine-tuned *ESM2* checkpoint (updated only via substrate-based kinase group classification), the resulting projections displayed improved clustering by group. These qualitative trends were confirmed with unsupervised clustering metrics, including the silhouette score and

the Calinski-Harabasz index. As shown in Figures 6 and 7 and Table 20, the fine-tuned encoder demonstrates clearer group structure despite never directly observing kinase sequences during training—suggesting that supervised signals from substrates alone can reorganize the encoder's representation space in a biologically meaningful way.

### A.4.2 REGRESSION

In the regression experiments, we fixed the majority of the *ESM* encoder parameters, unfreezing only the last six layers for joint fine-tuning with the decoder. For comparison, baseline models attach a single linear regression head to a frozen encoder. Because Spearman correlation is invariant to monotonic transformations, we found that min–max scaling the labels to $[0, 1]$ with four digits precision after floating point markedly improved convergence and performance: the decoder learns the structure of numerical outputs more rapidly when they occupy a consistent range. For the RMSE report, we scaled the numbers back to their original range.

To evaluate sequence-based prediction of structural similarity, we tokenized the ProteinShake structure-similarity dataset (*Structure Split*) and concatenated each pair of sequences with an <EOS> separator. Only the last four encoder blocks were trainable, and batches contained 128 sequence pairs. Results are summarised in Table 18.

Table 21: Protein–protein structure similarity on the ProteinShake test set (*Structure Split*). All ProteinShake baselines rely on 3-D structural input; † denotes a linear layer fine-tuned on the last four encoder blocks.

| Method | Spearman $\rho$ |
|---|---|
| Baseline (ESM-2†) | 0.4653 |
| ProteinShake (Graph) (Kucera et al., 2024) | 0.5180 |
| ProteinShake (Point) (Kucera et al., 2024) | 0.5640 |
| ProteinShake (Voxel) (Kucera et al., 2024) | 0.5730 |
| Prot2Token-C | 0.5267 |

Regarding the protein-protein affinity task, the labels were normalized to $[0, 1]$. We used the same hyperparameters of structure-similarity task, with a freshly initialized decoder. Performance, reported as RMSE (lower is better), appears in Table 21.

Table 22: Protein–protein binding affinity prediction on PPB-Affinity. The baseline is based on ESM-2 650m encoder.

| Method | RMSE ($\downarrow$) |
|---|---|
| baseline (Liu et al., 2024) | 2.1040 |
| Prot2Token-C | 1.6632 |

For the HotProtein HP-S regression split we applied the same min–max label normalisation as in the other regression tasks. Results are reported in Table 22, improving upon the HotProtein by roughly 2.5–3.0 pp in both correlation metrics.

Table 23: Performance of predicting thermostability on HotProtein (HP-S test split, fold 1). † denotes a linear layer fine-tuned on the last four encoder blocks.

| Method | Spearman | Pearson |
|---|---|---|
| TAPE (Rao et al., 2019) | 0.504 | 0.453 |
| ESM-1B | 0.807 | 0.809 |
| HotProtein (Chen & Gong, 2022) | 0.823 | 0.827 |
| Prot2Token-C | 0.8437 | 0.8439 |
| Baseline (ESM-2†) | 0.8644 | 0.8699 |

### A.4.3 SELF-SUPERVISED PRE-TRAINING OF DECODER

In our preliminary experiments with the phosphorylation PTMs and protein-ligand binding site tasks, we initially focused on directly predicting positive sites using the *Prot2Token* framework. However,

we observed suboptimal performance despite experimenting with different label formatting methods. Upon further analysis, we hypothesized that this issue arose primarily from the lack of inductive biases inherent to specialized models, which were missing in the *Prot2Token* model's decoder due to its random initialization.

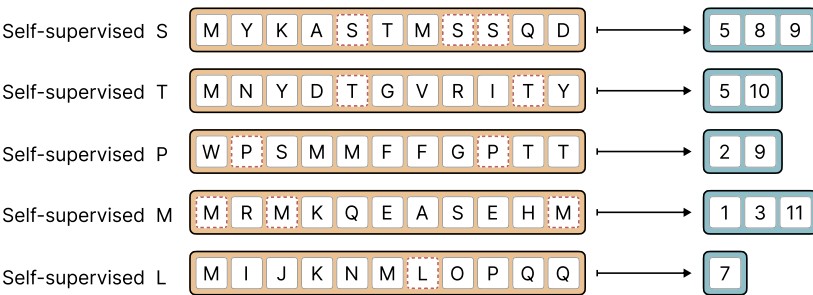

Figure 8: Illustration of self-supervised pre-training tasks designed for the decoder. For each amino acid (e.g., `S, T, P, M, L`), the model is trained to predict the positions of its occurrences within a given protein sequence. Highlighted residues are the targets, and the output is a list of their corresponding indices. This enables the decoder to learn position-aware amino acid representations in a label-free manner.

Specialized baseline approaches commonly restrict the prediction space by focusing on specific amino acids known to undergo modifications, such as serine (`S`), threonine (`T`), and tyrosine (`Y`) in phosphorylation tasks. These approaches implicitly encode biases about label structures into their predictive mechanisms. Conversely, *Prot2Token*, being an autoregressive model with a randomly initialized decoder, initially lacked these intrinsic biases, severely impacting its predictive accuracy, especially in tasks with extensive label vocabularies.

To address this challenge, we introduced a self-supervised pre-training strategy to effectively embed inductive biases into the decoder before fine-tuning it on the main supervised tasks. The key idea behind this self-supervised approach is straightforward yet effective: the decoder is trained to recognize amino acid positions within sequences (Figure 8). For instance, given an amino acid sequence such as `MSGLSNYT`, the model learns to output positional indices 2, 5 corresponding to the amino acid `S`. We constructed twenty such self-supervised tasks, each dedicated to recognizing the positions of a different amino acid type. Importantly, generating these self-supervised samples does not require human annotation, making it a cost-effective method to improve model initialization and predictive performance.

Our empirical results, presented in Table 23, demonstrate a clear positive correlation between the volume of self-supervised auxiliary samples and model performance improvements on the phosphorylation task. Notably, incorporating a broader range of amino acids, such as `K`, `N`, and `R`, alongside the typical phosphorylation targets (`S, T, Y`), significantly boosted model accuracy, highlighting the utility of teaching these biases to the decoder.

Table 24: Phosphorylation site prediction. "Aux" denotes self-supervised auxiliary tasks. All results are based on *Prot2Token-B* model.

| Data | Accuracy | $F_1$ |
|------|----------|-------|
| Phosphorylation | 55.69 | 0.0198 |
| Phosphorylation + STY-Aux (150k) | 74.57 | 0.0592 |
| Phosphorylation + STY-Aux (250k) | 91.49 | 0.1799 |
| Phosphorylation + STYKNR-Aux (250k) | 94.14 | 0.3052 |

Furthermore, Figures 9 and 10 illustrate the frequency distribution of labels in phosphorylation and protein-ligand binding site tasks, respectively. These figures clearly show the imbalanced and

sparse nature of labels, underscoring why explicit inductive biases provided through self-supervised pre-training are crucial for effective model training.

We pre-trained the decoder once using 20 self-supervised tasks—each targeting the positional prediction of one amino acid type—to serve as a general-purpose initialization for all downstream tasks involving binding site prediction. This avoids the computational cost of re-adding auxiliary self-supervised tasks per downstream task, while still equipping the decoder with biologically meaningful priors.

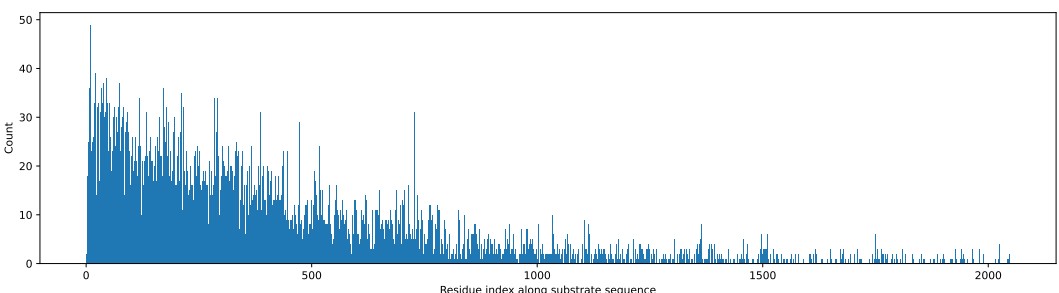

Figure 9: Distribution of phosphorylation-site indices in the training set (n = 11,449 sites across 5,694 peptides). Only residues at positions ≤ 2048 are shown; 176 rarer sites at higher indices were excluded. Each bar corresponds to a single residue position (bin width = 1).

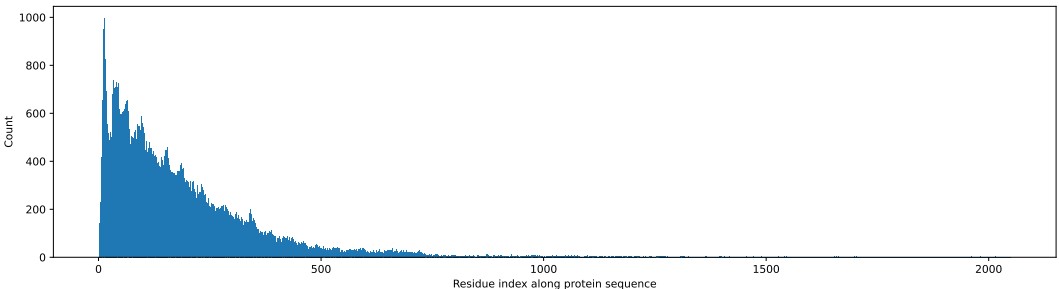

Figure 10: Distribution of protein–ligand binding-site indices aggregated across all 41 ligand classes in the training set. Bars represent individual residue positions (`bin width=1`). Sites located beyond residue 2048 (< 2 % of all annotated positions) were excluded for clarity.

A critical consideration in applying this self-supervised learning strategy is maintaining a frozen encoder during the pre-training phase. Allowing updates to the encoder parameters at this stage can inadvertently introduce shortcut learning effects, causing the model to collapse and diminishing its predictive capabilities on downstream tasks. Consequently, freezing the encoder ensures that the decoder robustly learns meaningful positional and structural biases, significantly enhancing its predictive performance on binding site types of tasks.

We randomly sampled 4 million protein sequences from the UniRef50 database (Suzek et al., 2015) for training and 4k for validation data. From them, we artificially created 80 million and 20k self-supervised samples, subsequently, by crafting each amino-acid-type/protein as one sample. Again, we sampled 1 million and 1k samples from them, respectively, to build the training and validation sets.

We used input sequence length of 2048, a weight decay of 0.01, and batch size of 192 samples, equivalent to 73,728 tokens. Also, we set the warmup steps to 512. We only froze the encoder weights and made other parameters trainable. After training for 16 epochs, the model showed perplexity of

2.31 on the validation set. This indicates that it almost perfectly converted the embeddings from the encoder back to their original protein sequences by learning to find the locations of each type of amino acid.

### A.4.4 BINDING SITE

Based on the order of ligands presented in Table 24, we grouped the ligands into distinguishable sets of 10, 20, 30, and all 41 ligands. Each ligand in a set was treated as a separate task defined by a task token figure 11, and trained together in one training session.

We selected a discrete task-token representation for ligands, rather than a continuous chemical encoder input (e.g., via SMILES), based on three key considerations. First, empirical evaluations demonstrated that our token-based approach yielded superior performance compared to architectures utilizing current pre-trained chemical encoders, suggesting limitations in the efficacy of chemical encoders for binding residue prediction. Second, this discrete formulation facilitates deep interpretability; by isolating ligand representations, we were able to analyze the learned embedding space to confirm that the model captures meaningful physicochemical relationships between ligands (A.5). Finally, a fixed vocabulary remains highly practical for many applications, such as ion binding, where the target space is naturally bounded, allowing our framework to robustly support a wider range of simultaneous targets (41 types) than many existing specialized predictors.

We selected each of those sets and jointly trained them alongside 20 self-supervised tasks using the latest checkpoint from the self-supervised pre-training phase. For this fine-tuning phase, the self-supervised tasks were reduced to a total number of 20k samples. Also, we removed protein samples with lengths greater than 1280 and set batch size to 98,304 tokens. During all training processes, only the last eight blocks of the encoder (*ESM2-650m*) were fine-tuned, while all non-encoder parameters of the supermodel were fully fine-tuned.

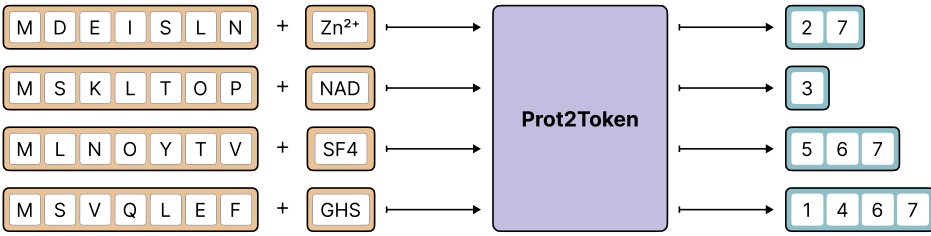

Figure 11: Jointly training protein–ligand binding-site across 41 types of ligands by representing ligands with task tokens.

It is worth noting that while we could have excluded the self-supervised tasks entirely from the fine-tuning stage, retaining a portion of these samples resulted in a measurable improvement in the model's performance on the supervised protein-ligand tasks.

Direct comparison of our method with other available methods was not straightforward due to several technical issues and potential overlap between their training data and our test sets; however, results of the comparison are provided in Table 25.

For fine-tuning on the protein-ligand datasets, the model was trained on a combined training set of selected ligands. During training, validation was performed for each ligand individually, and the best checkpoint for each ligand was saved based on its validation set performance. At the end of training, these best checkpoints were evaluated on their respective test sets. Figure 12 shows the average validation $F_1$ score across epochs, with the highest average performance observed at epoch 30. However, this checkpoint showed slightly lower average test performance compared to using individual best checkpoints for each ligand.

The results for all ligands are presented in Table 25. To compute the metric for the autoregressive model's output, each amino acid in a protein was treated as an individual positive or negative sample. Predicted binding residues from the decoder were considered positive samples, while all other amino

acids were treated as negative (zero) samples. The metrics were then calculated based on this classification.

To provide a comparison of our model's performance with other available methods, we present the results in Table 26. However, the comparison process faced several challenges: some web servers were not operational during testing, while others only allowed predictions on individual samples, making bulk evaluation difficult and very slow to response. We attempted to evaluate IonCom (Hu et al., 2016), and MIB2 (Lu et al., 2022) server tools, but encountered several issues: MIB2 had extremely slow response times, and IonCom imposed strict sample limitations for evaluation.

Table 25: $F_1$ scores of all ligands across different training configurations, with varying numbers of auxiliary ligands on the test sets. The table summarizes the impact of jointly training with 10, 20, 30, and 41 ligands on binding site prediction. * indicates that pre-trained decoder weights were not used, and † indicates that self-supervised tasks were excluded during supervised training.

| Ligands | 10 tasks †* | 10 tasks* | 10 tasks | 20 tasks | 30 tasks | 41 tasks |
|---|---|---|---|---|---|---|
| $ZN^{2+}$ | 0.0678 | 0.0657 | 0.7434 | 0.7498 | 0.7594 | 0.7575 |
| $CO^{2+}$ | 0.1022 | 0.0888 | 0.6566 | 0.6493 | 0.6472 | 0.6474 |
| CLA | 0.2749 | 0.2519 | 0.477 | 0.3763 | 0.4936 | 0.4762 |
| FAD | 0.1744 | 0.1476 | 0.6882 | 0.6565 | 0.6473 | 0.6537 |
| HEM | 0.243 | 0.232 | 0.6554 | 0.6698 | 0.6871 | 0.6796 |
| NAD | 0.1662 | 0.1248 | 0.6862 | 0.6851 | 0.6862 | 0.6952 |
| ADP | 0.1105 | 0.1053 | 0.6057 | 0.5779 | 0.5897 | 0.5834 |
| $MG^{2+}$ | 0.482 | 0.0326 | 0.4466 | 0.4603 | 0.4522 | 0.4575 |
| NAP | 0.1559 | 0.1417 | 0.6629 | 0.6813 | 0.6861 | 0.6746 |
| ATP | 0.1059 | 0.0927 | 0.4538 | 0.4355 | 0.5317 | 0.505 |
| **Average (top 10)** | **0.1883** | **0.1283** | **0.6076** | **0.5942** | **0.6181** | **0.6130** |
| HEC | - | - | - | 0.6438 | 0.6511 | 0.6537 |
| SF4 | - | - | - | 0.6508 | 0.584 | 0.5685 |
| FMN | - | - | - | 0.6921 | 0.6983 | 0.6945 |
| SAH | - | - | - | 0.6385 | 0.6473 | 0.6503 |
| NDP | - | - | - | 0.7122 | 0.7085 | 0.6979 |
| ANP | - | - | - | 0.6153 | 0.6214 | 0.6217 |
| GDP | - | - | - | 0.5948 | 0.6335 | 0.6465 |
| GLC | - | - | - | 0.2091 | 0.2237 | 0.2214 |
| PLP | - | - | - | 0.777 | 0.778 | 0.762 |
| $MN^{2+}$ | - | - | - | 0.7278 | 0.7245 | 0.7376 |
| **Average (top 20)** | - | - | - | **0.6102** | **0.6172** | **0.6130** |
| COA | - | - | - | - | 0.3978 | 0.4011 |
| SAM | - | - | - | - | 0.6355 | 0.6252 |
| AMP | - | - | - | - | 0.4316 | 0.4432 |
| BGC | - | - | - | - | 0.2165 | 0.1932 |
| $FE^{3+}$ | - | - | - | - | 0.6756 | 0.6606 |
| MAN | - | - | - | - | 0.1407 | 0.1216 |
| FES | - | - | - | - | 0.7162 | 0.7018 |
| $PO_4^{3-}$ | - | - | - | - | 0.2288 | 0.2278 |
| GTP | - | - | - | - | 0.5332 | 0.5461 |
| UDP | - | - | - | - | 0.5522 | 0.5391 |
| **Average (top 30)** | - | - | - | - | **0.566** | **0.5615** |
| $CU^{2+}$ | - | - | - | - | - | 0.5607 |
| GSH | - | - | - | - | - | 0.6924 |
| AGS | - | - | - | - | - | 0.5301 |
| ACO | - | - | - | - | - | 0.5026 |
| GAL | - | - | - | - | - | 0.2762 |
| $SO_4^{2-}$ | - | - | - | - | - | 0.1386 |
| CLR | - | - | - | - | - | 0.0373 |
| Y01 | - | - | - | - | - | 0.0419 |
| BMA | - | - | - | - | - | 0.2273 |
| $FE^{2+}$ | - | - | - | - | - | 0.6033 |
| $CO^{2+}$ | - | - | - | - | - | 0.517 |
| **Average (all)** | - | - | - | - | - | **0.5115** |

Additionally, a potential overlap between the training data of these methods and our crafted test sets further made a fair evaluation complicated. This was particularly evident for LMetalSite (Yuan et al., 2022a), where their reported performance on their own test sets was significantly lower compared to their results on our test sets, indicating a sign of data leaking in this comparison.

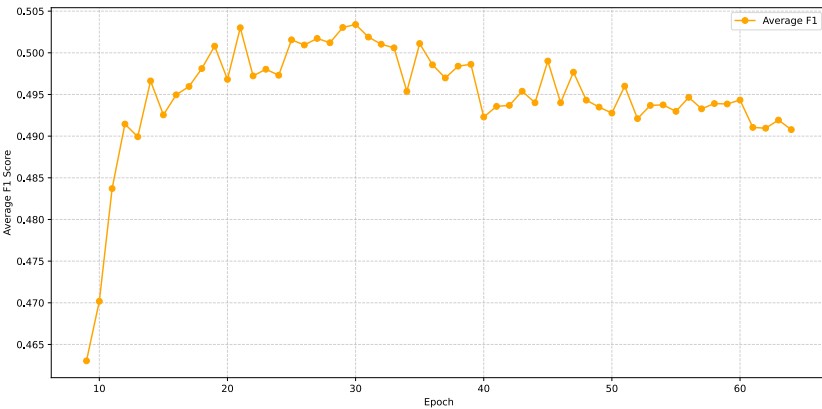

Figure 12: Average of $F_1$ values for all 41 ligands during the training based on the validation sets. The performance peaked at epoch 30.

Finally, preliminary experiments were conducted on Protein-Protein Binding Site Prediction using the PPIRef dataset (Bushuiev et al.). The task involved predicting the binding interface residues on a target protein given the sequences of both the target and binder proteins (concatenated as input). Binding residues were defined as those on the target protein within 6Å of the binder. Initial results using the simplified index tokenization yielded an $F_1$ score of 0.47 on the test set. While encouraging, this result is preliminary, and further investigation is required.

Table 26: Comparison of our method's best performance for each ligand with other available methods on selected ligands based on $F_1$ score. The main values are based on their reported test set performance as described in their respective papers. * Indicates they are reported on our test sets.

| Ligand | Metrics | Prot2Token-C | TargetS (Yu et al., 2013) | LMetalSite (Yuan et al., 2022b) | ZinCap (Essien et al., 2019) | MIB2 (Lu et al., 2022) |
|---|---|---|---|---|---|---|
| $CA^{2+}$ | $F_1$ | 0.6566* | 0.392* | 0.526 (0.7370*) | - | - |
| | MCC | - | 0.320 (0.431*) | 0.542 (0.7342*) | - | - |
| | Acc | - | 0.984 (0.977*) | 0.9884* | - | 0.941 |
| $MG^{2+}$ | $F_1$ | 0.4603* | 0.433* | 0.367 (0.5560*) | - | - |
| | MCC | - | 0.383 (0.450*) | 0.419 (0.5773*) | - | - |
| | Acc | - | 0.990 (0.992*) | 0.9949* | - | 0.946 |
| $ZN^{2+}$ | $F_1$ | 0.7594* | 0.660* | 0.76 (0.8299*) | 0.451* | - |
| | MCC | - | 0.557 (0.660*) | 0.761 (0.8275*) | 0.54 (0.48*) | - |
| | Acc | - | 0.989 (0.989*) | 0.9953* | 0.870 (0.97*) | 0.948 |
| $MN^{2+}$ | $F_1$ | 0.7376* | 0.579* | 0.662 (0.8048*) | - | - |
| | MCC | - | 0.445 (0.574*) | 0.661 (0.8024*) | - | - |
| | Acc | - | 0.987 (0.989*) | 0.995* | - | 0.950 |

### A.4.5 SEQUENCE-TO-SEQUENCE

For the secondary structure prediction task, *Prot2Token* was trained to assign a secondary structure class to each residue in the input protein sequence, treating the problem as a sequence-to-sequence token prediction. The dataset was preprocessed to map residues to standard secondary structure labels (helix, sheet, coil). Performance was evaluated using the macro-$F_1$ score on the test set of PEER. As shown in Table 27.

For the sequence-to-3D structure prediction task, we fine-tuned the last six blocks of the *ESM2-650m* encoder within the *Prot2Token* framework. We used 8192 warmup steps for this particular task. The model was trained to generate discrete structure tokens corresponding to backbone coordinates, utilizing a VQ-VAE-based tokenizer. The current VQ-VAE implementation supports protein sequences in the range of 50 to 512 residues. During training, model performance was evaluated using TM-score on the test set explained in A.3.7, and at the end, the best checkpoint is compared with other methods on a subset of CAMEO dataset reported in A.3.7.

During inference, we encountered a challenge where the decoder occasionally generated an output sequence with either more or fewer tokens than the actual number of amino acids in the input sequence.

To address this issue, we applied a constraint on the end `<EOS>` token probability. Specifically, during inference, we artificially adjusted the probability of the `<EOS>` token, ensuring that it was only allowed if the number of predicted 3D tokens matched the length of the input amino acid sequence. This adjustment effectively enforced sequence alignment and resolved inconsistencies in output length of generated structure.

Table 27: Secondary structure prediction evaluation. The baseline involves a linear classifier on top of the frozen *ESM* model.

| Method | Macro-$F_1$ | Model |
| --- | --- | --- |
| PEER (Xu et al., 2022) (fine-tuned) | 82.73 | ESM1-1b |
| Baseline | 84.78 | ESM2-650m |
| Prot2Token-B | 83.56 | ESM2-650m |

To further evaluate the learned structure-aware representations, we utilized the CATH-labeled protein sequences from (Wang et al., 2025a), specifically the `CATH_nonredundant_S40` dataset (release v4_3_0). In this dataset, no two sequences share more than 40% identity, and one representative (the longest) from each CATH superfamily is selected. This provides a challenging testbed for assessing the structural awareness of protein embeddings across three hierarchical CATH levels: Class, Architecture, and Topology.

In addition to structural benchmarks, we examined functional grouping using kinase and deaminase family datasets. Kinase domain sequences and their group labels were extracted from GPS 5.0 (Wang et al., 2020), resulting in 336 kinases from nine groups. Deaminase sequences and their respective family annotations were curated from a reference dataset (Huang et al., 2023). For each protein, we generated embeddings and assessed whether these could successfully separate functional classes.

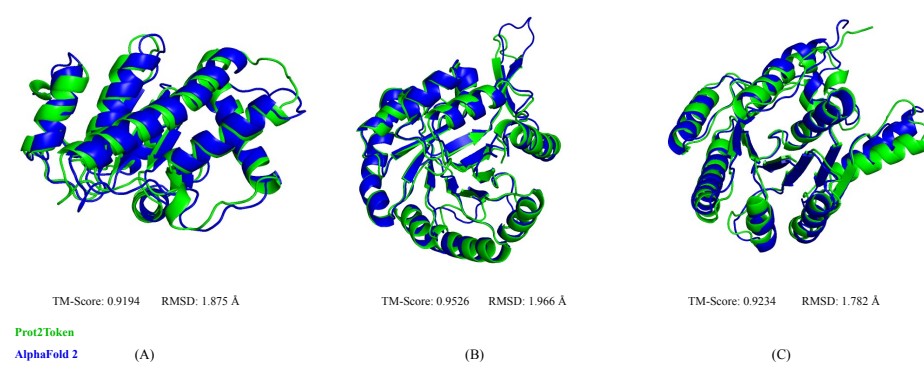

Figure 13: Randomly selected test set samples where our model achieved a TM-score above 0.90 versus *AF2* high-pLDDT predictions. On average, each sample was predicted and converted in approximately 1 second using an Nvidia A100 GPU.

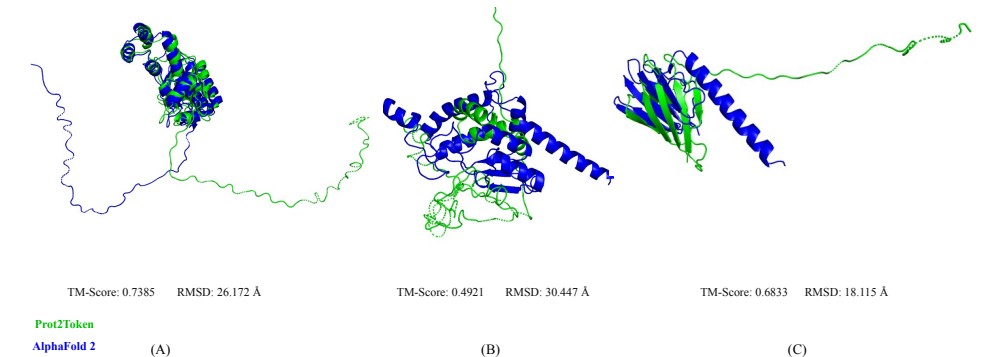

Figure 14: Randomly selected test set samples where the model achieved a TM-score lower than 0.75.

Table 28: Evaluation of protein embedding quality via clustering. Clustering performance is reported for CATH structural hierarchy levels using the Calinski-Harabasz index (CHI), and for functional groupings (kinase, deaminase) using Adjusted Rand Index (ARI). Higher values indicate more accurate and biologically meaningful clusters. *S-ESM* stands for structure-aware *ESM*.

| Methods | CATH (CHI) | | | Kinase group clustering (ARI) | Deaminase clustering (ARI) |
|---|---|---|---|---|---|
| | Class | Architecture | Topology | | |
| ESMC-600m (ESM Team, 2024) | 19.87 | 7.70 | 4.06 | 0.1720 | 0.4067 |
| ESM2-650m | 13.16 | 7.30 | 4.35 | 0.2691 | 0.6473 |
| S-ESM (Prot2Token-C encoder) | 44.40 | 19.34 | 11.50 | 0.5806 | 0.7963 |

Clustering quality for the functional groups (kinase and deaminase families) was quantified using the Adjusted Rand Index (ARI) after K-means clustering, while clustering for CATH structural categories (Class, Architecture, Topology) was measured by the Calinski-Harabasz Index (CHI) (Caliński & Harabasz, 1974), which captures the ratio of between- to within-cluster dispersion. Table 28 summarizes the results for all models. *Prot2Token*'s encoder achieves markedly higher CHI and ARI scores, especially in clustering kinase (ARI = 0.5806) and deaminase families (ARI = 0.7963), indicating improved capture of both structural and functional organization.

For qualitative comparison, Figure 16 presents a t-SNE visualization of protein embeddings colored by true structural or functional labels. Compared to *ESM2-650m* and *ESMC-600m*, *Prot2Token* embeddings yield more distinct and interpretable clusters that align with biological classification, demonstrating both stronger structural feature extraction and functional group separation.

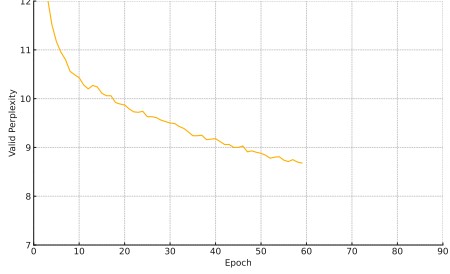

Figure 15: Validation perplexity curve for sequence-to-3D structure prediction. While perplexity keeps decreasing, the validation TM-score saturates at ≈0.55 on CAMEO 2024. Post-hoc analysis shows the 3D tokenizer (VQ-VAE) itself reconstructs with an upper bound of ≈0.60 TM-score on the same benchmark; hence the plateau reflects a tokenizer-imposed ceiling rather than insufficient optimization of *Prot2Token*. Improving the tokenizer is likely required to push beyond this regime.

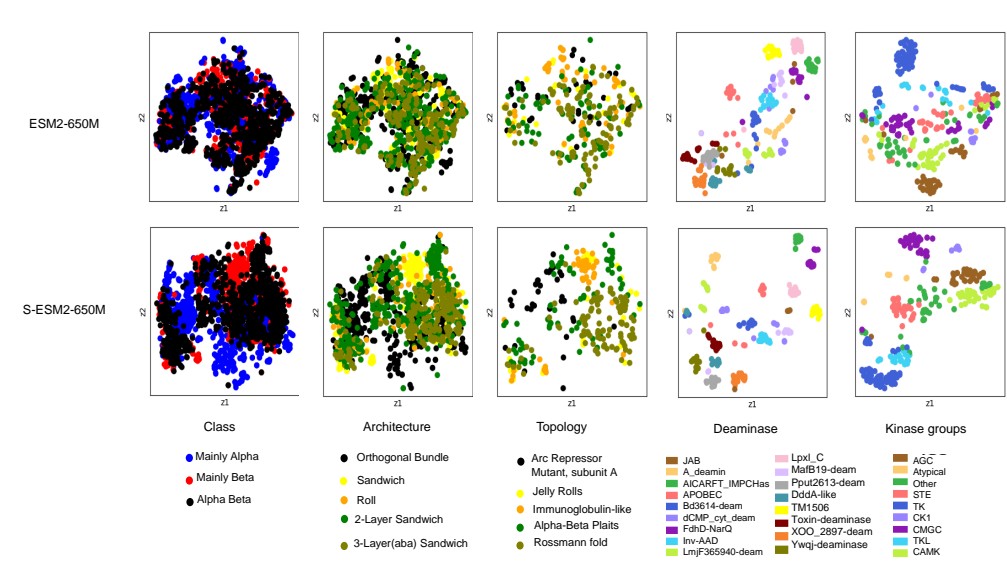

Figure 16: Comparison of protein representations generated by *Prot2Token* and the base encoder *ESM2-650m*. The t-SNE visualizations display protein embeddings for CATH structural classes, architecture, and topology, as well as clustering for deaminase families and kinase groups. Colors correspond to the known classes or families for each category.

### A.4.6 OTHER

We utilized the TargetP-2 dataset which encompasses both cleavage site data and five types of localization labels. We represented the label format as a combination of classification and regression tasks, for instance, {sp, 96}, where sp denotes the localization label (Signal Peptide) and 96 indicates the cleavage site's location. Additionally, to evaluate the model, we implemented a 5-fold cross-validation strategy. We considered fine-tuning only the last layer of the *ESM* models for both the *Prot2Token* model and the baseline comparison. Table 15 presents a comparative analysis of *Prot2Token* against *ESM* with a linear classifier head. The results suggest that by enabling the model to learn the locations of different amino acids through self-supervised auxiliary tasks, it achieves more accurate predictions of cleavage site positions. Furthermore, the performance in localization prediction also shows improvement with the integration of auxiliary tasks. We attribute this enhancement in performance to the model's improved understanding of cleavage site positions. Note that the performance of bigger models was very similar to the smaller ones.

Table 29: Localization and cleavage site prediction. "Aux" denotes self-supervised auxiliary tasks using STYKNR amino acids. Localization and cleavage site metrics are based on Macro-$F_1$ and MAE, respectively.

| Method | Aux-Tasks | Cleavage Site | localization |
|---|---|---|---|
| Baseline (ESM2-35m) | - | - | 90.96 |
| Prot2Token-A | - | 3.6392 | 90.56 |
| Prot2Token-A | Aux (12k) | 2.9205 | 92.30 |

In the next step, we fine-tuned the model starting from the latest checkpoint obtained during the self-supervised pre-training stage that is reported in Appendix A.4.3. This process involved jointly training six PTMs alongside self-supervised samples. The maximum sequence length for input protein sequences was set to 1024 tokens, and the batch size was adjusted to process 98,304 tokens per iteration.

Notably, while it was possible to exclude self-supervised tasks entirely during fine-tuning, retaining a subset of these samples led to improved generalization and enhanced performance on the protein-kinase phosphorylation site prediction. From the 33 total blocks in the protein encoder, we selectively fine-tuned the last eight blocks by unfreezing their weights for training. The results are presented in Table 30.

Table 30: PTMs comparison based on $F_1$ score on our test sets. $^\dagger$: There is a strong possibility of data contamination between our test set and the PTMGPT2 training set. As a result, PTMGPT2 may achieve artificially high performance on our test set due to memorization, while its real-world performance on unseen samples could be lower.

| PTM | *Prot2Token* (Ours) | ESM-2 | PTMGPT2$^\dagger$ (Shrestha et al., 2024) |
|---|---|---|---|
| Ubiquitylation | 0.1382 | 0.1993 | 0.165 |
| Phosphorylation | 0.4055 | 0.3908 | 0.400 |
| Acetylation | 0.307 | 0.3273 | 0.350 |
| Methylation | 0.4608 | 0.4532 | 0.596 |
| N-linked Glycosylation | 0.9689 | 0.9586 | 0.862 |
| O-linked Glycosylation | 0.4695 | 0.4597 | 0.531 |
| Succinylation | 0.2663 | 0.3515 | 0.540 |

### A.5 PROTEIN-LIGAND BINDING SITE TASK TOKENS INTERPRETATION

In this section, we scrutinized the task token embeddings of the decoder that was pre-trained on all 41 ligands in the previous section to find the sign of chemical properties of ligands and their relationships together.

Empirically, based on the $F_1$ scores of the ligands that the model was trained and evaluated on, the task token embeddings successfully captured meaningful representations of the ligands. However, to solidify this framework as a foundation for future research, we aimed to validate these embeddings from an additional perspective. Our goal is to create a robust infrastructure that can incorporate more ligands into a single model, thereby addressing the scarcity of data for certain ligands through knowledge transfer between ligands. To achieve this, first, from all 41 ligands, we selected top 28 ligands based on $F_1$ score and filtered the rest and then, we analyzed the task token embeddings of the remaining ligands by clustering them to explore ligand similarities in the embedding space.

Simultaneously, we clustered the ligands based on their biochemical features in the real world, and in the last step, we investigated the correlation between these two clustering approaches. The purpose of this comparison was to determine whether the learned task token embeddings genuinely reflect real-world relationships between ligands or if they merely memorize specific patterns without capturing meaningful biochemical similarities. Figure 17 highlights the intersection between the two spaces of ligand representations: the embedding space and the biochemical feature space. It illustrates which ligands or sets of ligands have their relationships successfully captured by the generated task token embeddings, as reflected by their agreement with relationships derived from biochemical features, and which embeddings failed to capture such relationships. More details, including feature selection, methods, and the interpretation algorithm are in the following subsections.

### A.5.1 INTERPRETATION

In this study, we developed a protein binding site prediction model using a multi-task learning framework, where each task represents a specific ligand. A 640-dimensional task token was incorporated for each ligand alongside the protein sequences. During training, the model learned meaningful task token embeddings that effectively represent ligands and their unique characteristics.

To validate the task token embeddings, we employed two clustering approaches: one based on the trained task token embeddings and the other on biochemical ligand features. For precise clustering and clearer analysis, ligands with an $F_1$ score below 0.5 were excluded to minimize noise, leaving 28 out of 41 ligands for analysis. Task token embeddings were reduced to 27 principal components using PCA, preserving 99% of the variance, and clustered with k-means to generate target clusters. For validating all ligands, the full set of 41 ligands was included. In this case, task token embeddings were reduced to 40 components to preserve 99% of the variance, and the same clustering method was applied. For the ligand features, 26 biochemical descriptors were collected, covering physical, chemical, electronic, hydrophilic, lipophilic, and geometric properties.

A systematic feature selection process evaluated all possible combinations of up to 13 features selected from these 26 descriptors (approximately 39 million combinations) to optimize clustering quality against the target clusters. The ARI was used as the selection metric, while Normalized Mutual Information (NMI) and Pairwise Accuracy metrics were later employed to evaluate the final selection.

The clustering results demonstrate that the learned task token embeddings are meaningful, as their clustering aligns closely with that based on ligand-specific biochemical features. Moderate-to-high agreement metrics (`ARI=0.447`, `NMI=0.614`, and `pairwise-accuracy=0.783`) highlight the embeddings' ability to capture key biochemical characteristics of ligands. Chemically significant features, such as `MolecularWeight`, `NetCharge`, and `RotatableBonds`, identified as part of the optimal feature set, further reinforce the relevance of the embeddings. The overlap and similarity in ligand grouping across both clustering approaches validate the hypothesis that the task token embeddings effectively encode biologically and chemically meaningful information.

However, reducing task token embeddings or biochemical features to 2D for visualization causes significant information loss, making 2D clustering plots less informative (Figures 23 and 24). These findings emphasize the importance of preserving higher-dimensional information for accurate interpretation and highlight the value of task token embeddings in ligand characterization for protein binding site prediction. Figure 18 shows the embeddings-based clustering, while Figure 19 shows the features-based clustering, and Figure 17 illustrates the global, local, and no relationships between the two approaches of embeddings-based clustering and features-based clustering.

**Global relationships.** Figure 17 highlights the ligands that have been clustered correctly across and within both clustering approaches. For instance, in Cluster 3, the solid circles for ACO, ATP, FAD, GTP, NAD, and SAM ligands represent ligands that have been consistently clustered across and within the same clusters in both approaches. This indicates that the task token embeddings successfully capture their similarity with each other and with the rest of the ligands.

**Local relationships.** Figure 17 also depicts ligands that have been clustered correctly only within clusters in both clustering approaches. For example, the stars in cluster 3 for $FE^{2+}$ and $MN^{2+}$ indicate that these ligands are grouped but appear in different clusters across the two approaches. Nevertheless, the task token embeddings still manage to capture their similarity with each other, even if they fail to capture their similarity with other ligands.

**No relationships.** For some ligands, the task token embeddings fail to accurately capture their global or local relationships. This may be due to the ligand features collected not being entirely representative and requiring further refinement, or because the task token embeddings themselves need improvement. Figure 17 illustrates these `no relationships` using triangles; for instance, the HEM ligand has been grouped with different ligands across different clusters in both approaches.

For further investigation of the task token embeddings, we incorporated all 41 ligands into the clustering analysis. The metrics showed a notable drop: `ARI=0.259`, `NMI=0.333`, and `Pairwise-Accuracy=0.733`. This decrease was expected, as including task token embeddings for ligands with low $F_1$ scores introduced some misaligned clusters. However, a closer examination reveals that the embeddings still effectively capture the global and local relationships between most ligands. Figures 20 and 21 depict the embeddings-based clustering and features-based clustering, respectively, while Figure 22 illustrates the global, local, and no relationships across all 41 ligands. Notably, out of the 41 ligands, the task token embeddings successfully represented 21 ligands globally, 13 ligands locally, and misrepresented 7 ligands. These results indicate that the task token embeddings consistently demonstrate strong global and local relationships, effectively capturing biochemical similarities among ligands. This reinforces the conclusion that the model has learned meaningful representations, even for ligands with low $F_1$ scores.

### A.5.2 FEATURES POOL CREATION

The feature pool of 26 descriptors was carefully designed to capture the physical, chemical, and structural properties of ligands, making them particularly suitable for describing protein-ligand interactions. These features were selected using domain knowledge of protein-ligand interactions and their ability to explain binding phenomena effectively. Two primary sources were used to collect these features:

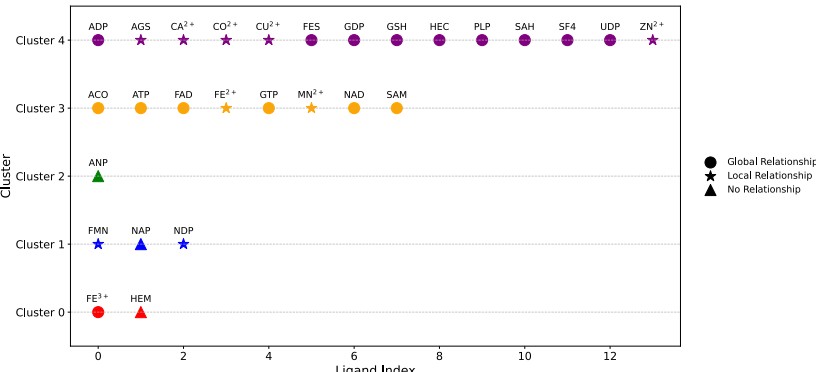

Figure 17: Global Relationships indicate that general biochemical features shared among many ligands have been captured. Local Relationships reflect the successful capture of biochemical properties between specific ligands and their closely related counterparts. No Relationships indicate that the biochemical properties were not captured at all.

- PubChem (Kim et al., 2023), a free online database maintained by the National Center for Biotechnology Information (NCBI), which provides precomputed chemical information for small molecules, drugs, and bioactive compounds. Features were retrieved using Compound IDs (CIDs) and *SMILES*, a text-based representation of molecular structures.

- The second source was RDKit, an open-source cheminformatics toolkit, where *SMILES* strings were converted into molecular objects and processed using various descriptors to compute additional features.

Table 31 shows the set of 26 features, categorized into seven groups, captures the properties of metal ions and molecules from multiple perspectives, providing a comprehensive description of their binding potential with proteins.

### A.5.3 OPTIMIZING FEATURE SELECTION

Our approach leverages clustered embeddings as a reference to evaluate clustered features from various feature combinations, identifying the best set of features to describe ligands based on the highest ARI score. We began with task token embeddings of ligands that achieved high $F_1$ scores to ensure noise reduction and high-quality clustering. These embeddings, initially 640-dimensional, were reduced to 27 principal components using PCA while retaining 99% of the variance. The reduced embeddings were then clustered using k-means, with the optimal number of clusters determined via the Elbow method, serving as the target clusters.

To identify the most informative ligand features, we implemented a search algorithm (Algorithm 1) that evaluates all possible combinations of up to 13 features from a pool of 26. In the first iteration, the algorithm selects a single feature (26 possible options). In the second iteration, it selects two features (325 possible combinations). This process continues up to 13 features, yielding approximately 39 million combinations. For each combination, the ligand-based feature clustering is performed, and the ARI score is computed. The feature combination that achieves the highest ARI score is selected as the best set.

Next, we removed the threshold constraint and extended the algorithm to all 41 ligands, examining whether task token embeddings captured meaningful representations for ligands with $F_1$ scores below 0.5. This analysis demonstrated that the embeddings retained significant information even for lower-performing ligands.

Table 32 presents the output of our searching algorithm, showing the top three feature combinations based on the ARI metric for the top 28 ligands. Table 33 displays the top three feature combinations for the entire set of 41 ligands.

Table 31: All 26 features we used in the interpretation step.

| No. | Name | Description | Source |
|---|---|---|---|
| 1 | MolecularWeight | Total molecular mass | PubChem |
| 2 | ExactMass | High-precision mass of the molecule | PubChem |
| 3 | MolecularVolume | Estimated molecular volume | RDKit |
| 4 | HeavyAtomCount | Count of non-hydrogen atoms | RDKit |
| 5 | RingCount | Total number of rings in the molecule | RDKit |
| 6 | CarbonCount | Number of carbon atoms | RDKit |
| 7 | OxygenCount | Number of oxygen atoms | RDKit |
| **Polarity and Hydrophobicity Features** | | | |
| 8 | LogP | Partition coefficient (hydrophobicity) | PubChem |
| 9 | MolLogP | Alternative measure of hydrophobicity | RDKit |
| 10 | HydrophobicSurfaceArea | Hydrophobic interaction area (TPSA) | RDKit |
| 11 | TPSA | Topological Polar Surface Area (polarity) | PubChem |
| 12 | Hydrophilicity | Difference between molecular weight and hydrophobicity | RDKit |
| 13 | Polarizability | Molecular refractivity | RDKit |
| 14 | Refractivity | A measure of a molecule's polarizability | RDKit |
| **Charge and Electrostatics Features** | | | |
| 15 | NetCharge | Net electrical charge of the molecule | PubChem |
| 16 | ElectrostaticPotential | Approximate measure of electrostatic potential | RDKit |
| **Flexibility and Rotational Features** | | | |
| 17 | RotatableBonds | Number of rotatable bonds | PubChem |
| 18 | RotatableBondFraction | Fraction of single bonds that are rotatable | RDKit |
| **Bond and Connectivity Features** | | | |
| 19 | SingleBonds | Count of single bonds | RDKit |
| 20 | DoubleBonds | Count of double bonds | RDKit |
| 21 | BalabanJ | Balaban index (topological descriptor) | RDKit |
| **Hydrogen Bonding Features** | | | |
| 22 | HBondDonors | Number of hydrogen bond donors | PubChem |
| 23 | HBondAcceptors | Number of hydrogen bond acceptors | PubChem |
| 24 | HydrogenBondingPotential | Difference between molecular weight and TPSA | RDKit |
| **Aromaticity and - Interactions** | | | |
| 25 | AromaticRings | Number of aromatic rings | RDKit |
| 26 | PiPiInteractionSites | Number of - interaction sites | RDKit |

---

**Algorithm 1** Ligand interpretation clustering

---

**Input** (features_pool (26 features), task_token_embeddings (640D), ligands (41), threshold (e.g., $F1 > 0.5$))
**Output** (best_features_combination, best_ari)
**Step 1. Preprocessing:**
  **(a) Filter Ligands:**
  high_quality_ligands $\leftarrow$ {ligand | $F1$(ligand) > threshold}
  **(b) Reduce Embeddings:**
  pca_embeddings $\leftarrow$ PCA(task_token_embeddings, $n$ components, 99% variance)
  **(c) Find Clusters:**
  $k_{\text{optimal}} \leftarrow$ ElbowMethod(pca_embeddings)
  target_clusters $\leftarrow$ KMeans(pca_embeddings, $k_{\text{optimal}}$)
**Step 2. Feature Combination Evaluation:**
  **Initialization:**
  best_ari $\leftarrow -\infty$
**for** $n_{\text{features}} = 1$ **to** 13 **do**
  combinations $\leftarrow$ Combinations(features_pool, $n_{\text{features}}$)
  **for each** combination $\in$ combinations **do**
    feature_clusters $\leftarrow$ KMeans(combinations, $k_{\text{optimal}}$)
    ari $\leftarrow$ ComputeARI(feature_clusters, target_clusters)
    **if** ari > best_ari **then**
      best_ari $\leftarrow$ ari
      best_features_combination $\leftarrow$ combination
    **end if**
  **end for**
**end for**

---

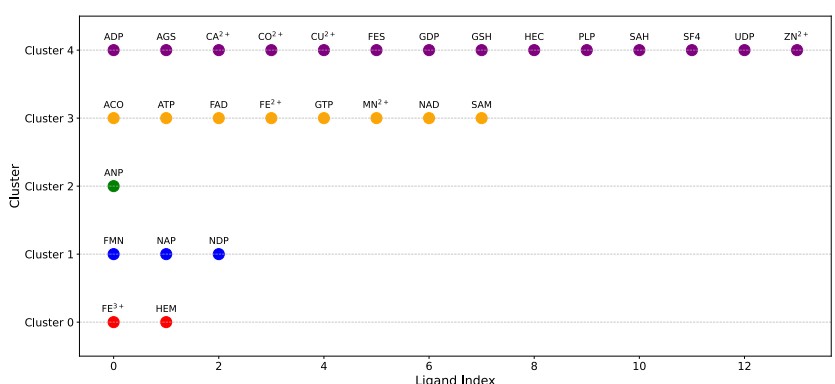

Figure 18: Clustering results of embeddings on top 28 ligands based on $F_1$ score.

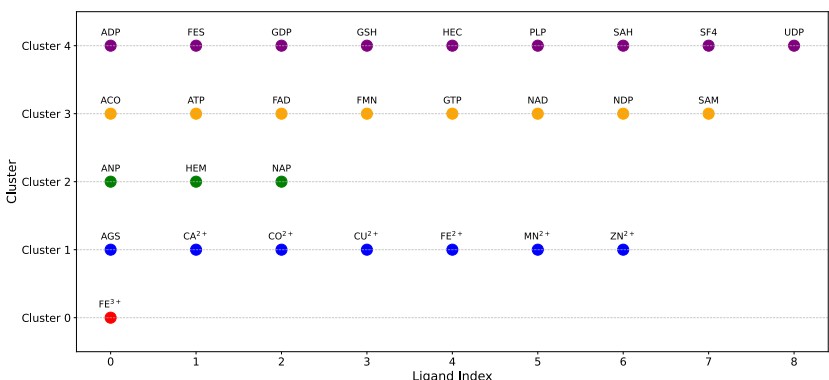

Figure 19: Clustering results of features on the 28 selected ligands.

### A.5.4 VISUALIZATION

To analyze the structural relationships within the high-dimensional ligand embeddings, we applied dimensionality reduction techniques to project the representation of 41 ligands from the 640 dimensional into a two-dimensional space for visualization. The methods explored included t-SNE (Figure 23) and UMAP (Figure 24).

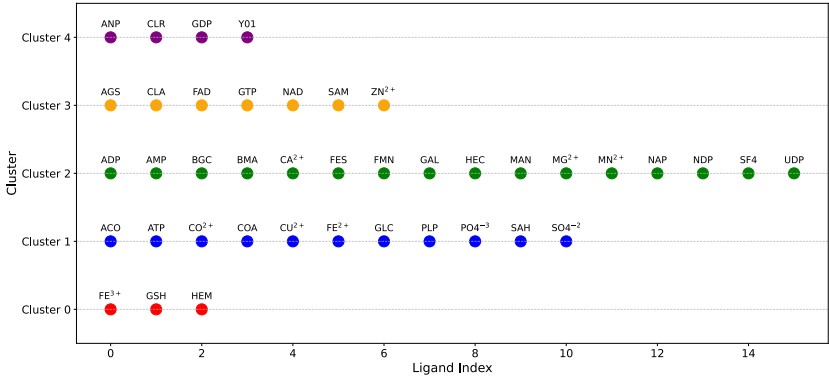

Figure 20: Clustering results of embeddings on all 41 ligands based on $F_1$ score.

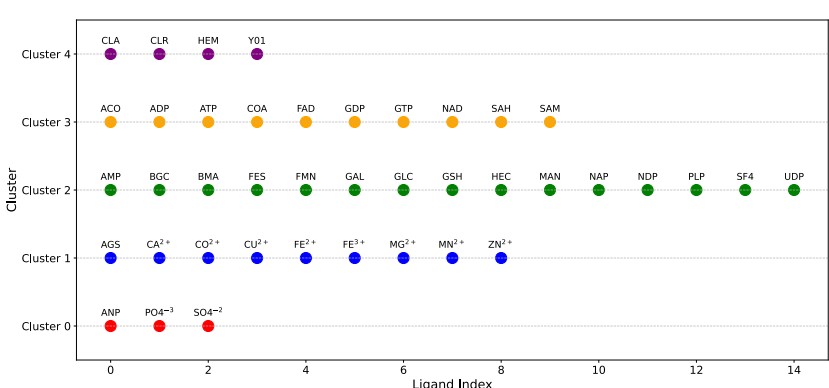

Figure 21: Clustering results of features on the 41 selected ligands.

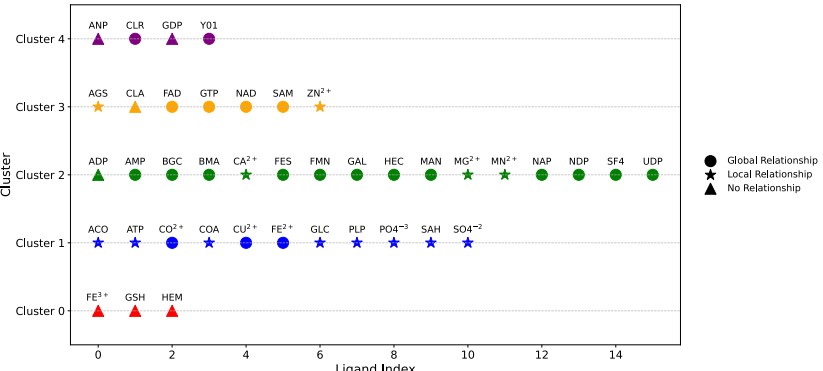

Figure 22: `Global Relationships` indicate that general biochemical features shared among many ligands have been captured. `Local Relationships` reflect the successful capture of biochemical properties between specific ligands and their closely related counterparts. `No Relationships` indicate that the biochemical properties were not captured at all.

Table 32: Top three feature combinations for the 28 ligands.

| No. Features | Features | ARI | NMI | Pairwise Accuracy |
|---|---|---|---|---|
| 7 | MolecularWeight, NetCharge, RotatableBonds, HydrogenBondingPotential, Carbon-Count, SingleBonds, BalabanJ | 0.447 | 0.614 | 0.783 |
| 12 | LogP, NetCharge, RotatableBonds, ExactMass, Polarizability, AromaticRings, Mol-LogP, MolecularVolume, HydrogenBondingPotential, CarbonCount, SingleBonds, BalabanJ | 0.423 | 0.573 | 0.772 |
| 13 | MolecularWeight, LogP, NetCharge, RotatableBonds, ExactMass, Polarizability, Mol-LogP, MolecularVolume, RingCount, HydrogenBondingPotential, CarbonCount, Bala-banJ, Hydrophilicity | 0.434 | 0.577 | 0.778 |

Table 33: Top three feature combinations for the entire set of 41 ligands.

| No. Features | Features | ARI | NMI | Pairwise Accuracy |
|---|---|---|---|---|
| 8 | NetCharge, HBondDonors, HBondAcceptors, ExactMass, Refractivity, Hydrogen-BondingPotential, SingleBonds, PiPiInteractionSites | 0.248 | 0.317 | 0.728 |
| 10 | MolecularWeight, LogP, RotatableBonds, TPSA, MolLogP, MolecularVolume, Single-Bonds, Hydrophilicity, ElectrostaticPotential, PiPiInteractionSites | 0.206 | 0.319 | 0.709 |
| 13 | LogP, NetCharge, HBondDonors, HBondAcceptors, TPSA, ExactMass, Polarizability, AromaticRings, Refractivity, DoubleBonds, BalabanJ, Hydrophilicity, PiPiInteraction-Sites | 0.259 | 0.333 | 0.733 |

The perplexity parameter for t-SNE was set to 3, and the number of neighbors (`n_neighbors`) for UMAP was also set to 3. These parameters were chosen to focus on capturing local relationships among ligand embeddings and to preserve some global structural details. Additionally, the dimensionality of the output was set to two (`n_components=2`) because the visualizations are in two dimensions. All other parameters were kept at their default settings.

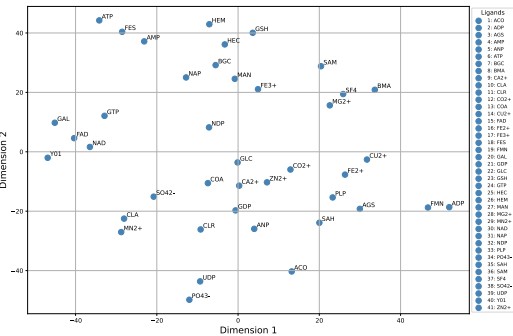

Figure 23: Visualization of task token embeddings using t-SNE.

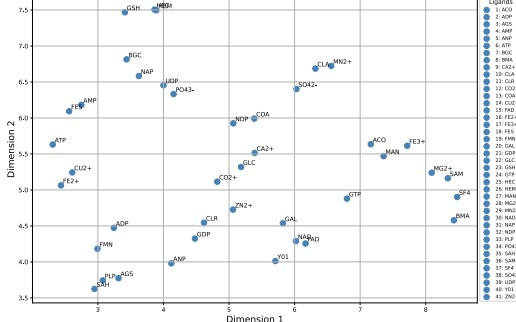

Figure 24: Visualization of task token embeddings using UMAP.

### A.5.5 MULTI-TASK LEARNING EFFECT

In this section, we tried to investigate the effect of multi-task learning on three ligands that have low number of samples but share similar semanticity in the embedding space of task tokens (Figure 17). Therefore, we selected GSH, CO, AGS as our target ligands because (1) they belong to the

same cluster, (2) showed global or local relationships and, (3) have less than 200 or lower number of protein sequences in total. We considered three groups to measure the performance of three ligands. There groups are defined as follow:

**Group 1.** Combination of the target ligands, GSH, CO and AGS (Equivalent to 1,819 tokens).

**Group 2.** Combination of CLA, FAD, HEM and NAD as ligands that did not share a close semantic representation in Figure 17 (Equivalent to $\sim 43k$ tokens).

**Group 3.** Combination of Zn, Ca, ADP, members from cluster 4 (Equivalent to $\sim 37k$ tokens).

Need to point out, in order to make the comparison of group 2 and 3 fair, we considered the total number of tokens in these groups close to each other. Table 34 shows that group 3 which shares a similar cluster with the target ligands, improves $F_1$ score more than other groups.

Table 34: The effect of jointly training under representative ligands based on different auxiliary groups with respect to $F_1$ score. "-" means no auxiliary task is used during training the target task.

| Target Task | - | Group 1 | Group 2 | Group 3 |
|---|---|---|---|---|
| GSH | 66.53 | 70.27 | 69.18 | 68.74 |
| CO | 35.87 | 32.65 | 29.26 | 58.48 |
| AGS | 31.63 | 22.94 | 43.18 | 49.09 |
| **Average** | **44.68** | **41.95 (-2.73)** | **47.21 (+2.53)** | **58.77 (+14.09)** |

## A.6 BROADER IMPACT

The *Prot2Token* framework represents a significant advancement in computational biology, with potentially transformative impacts on protein research, therapeutic discovery, and biotechnology applications. By unifying diverse protein prediction tasks within a single, scalable architecture, *Prot2Token* substantially reduces computational requirements and simplifies model management. This democratization of sophisticated predictive capabilities could significantly enhance accessibility for research groups with limited computational resources, facilitating broader participation and innovation in the field. Moreover, the substantial speed improvements demonstrated by *Prot2Token*, particularly in protein structure prediction, may enable real-time applications in clinical and industrial settings, such as personalized medicine, real-time drug screening, and rapid biomarker discovery.

However, alongside these benefits, the wide applicability and powerful predictive capacity of *Prot2Token* also necessitate careful ethical consideration. As the barrier to rapid protein prediction and generation lowers, it becomes increasingly important to implement responsible practices around data usage and sharing, ensuring that predictive outputs, especially those related to therapeutics or biologically active molecules, are validated rigorously before clinical or environmental deployment. Additionally, there is a need to consider the implications of such advanced modeling capabilities on biosecurity. With models that can rapidly predict or design biologically active proteins, safeguards must be established to prevent misuse, including unintentional production of harmful or disruptive biological agents. Continued dialogue and collaboration between computational biologists, policymakers, and ethicists will be crucial to navigating these challenges responsibly.

## A.7 LLM USAGE

In this manuscript, we used large language models only for copy-editing: improving grammar, clarity, and style of author-written text.

