# OpenReview forum: "Prot2Token: A Unified Framework for Protein Modeling via Next-Token Prediction"
_ICLR.cc/2026/Conference — Submitted to ICLR 2026_

### Official Review · Reviewer_6FgX · 2025-10-29

**Soundness:** 2
**Presentation:** 3
**Contribution:** 3
**Rating:** 4
**Confidence:** 4

**Summary:**

The authors present Prot2Token, an interesting approach to multitask learning for protein-related tasks. This is a difficult problem: the variety of tasks spans distinct input and output subspaces, so learning how to bring it all together is tricky. The authors went for a maximalist approach where every task becomes next-token prediction. The alternative would have been more custom prediction heads fed from shared encoder representations. I'm not convinced the next-token approach works well, and the evaluations (including some missing ablations) underscore this. The framework shows promise as a unified scaffold, but the "everything is auto-regressive token prediction" philosophy seems to create more problems than it solves.

Let me start with the ProteinGym evaluation, which is a red flag. I don't buy that they're outperforming a well-tested benchmark by this magnitude (Spearman $\rho=0.93$ vs. prior best $\sim 0.61$). This makes me think there's substantial data leakage happening. Could it be that the correlations aren't per-assay but are instead pooled? The latter would artificially inflate results through inter-assay signal.

This is compounded by the odd setup for ligand binding prediction. Most DTI tasks aren't designed to treat each ligand as a separate classification task. For metal ions, I could imagine that, but for most SMILES strings, the formulation should handle arbitrary chemical compounds as input. Their approach uses fixed task tokens for 41 specific ligands (Section A.4.4), making it impossible to generalize to new compounds.

These issues feed into my overall sense that cross-task learning hasn't been evaluated properly. The whole point is showing across-the-board benefits. While the authors found episodic benefits (training on tasks B, C, D helps task A), the right experiment would systematically evaluate: what happens if I train on everything except this task? They show cherry-picked combinations (Tables 2, 7) but no comprehensive task influence matrix or negative transfer analysis.

Even if these evaluations were fixed, my fundamental concerns remain about how expressive next-token prediction can be for diverse tasks. Binding site prediction via sorted indices is fragile; I saw no analysis where binding sites were provided in reordered settings. The authors acknowledge that tokenizing continuous values as ASCII characters is a choice and defend it, but it's still deeply unsatisfying-- they don't seem to have tried even a right-to-left tokenization? The decoder pretraining is clever but feels like a band-aid on a fundamentally problematic design.

Additional Concerns:

- Inconsistent architectural choices: Different tasks unfreeze different numbers of encoder layers (4, 6, or 8 blocks) without justification or ablation studies examining this crucial hyperparameter.

- Structure prediction quality gap: TM-score of $0.54$ vs. AF2's $0.88$ is presented as a speed win, but the model isn't usable for practical applications. If the VQ-VAE is bottleneck, maybe look at the tokenization of newer PLMs like ESM-3, which might offer better results?

- Missing principled baselines: Many of the baselines are simpler, less-strong ones rather than the strongest task-specific baselines around.

**Strengths:**

see above

**Weaknesses:**

see above

**Questions:**

see above

---

> ### Author Response · Authors · 2025-11-24
>
> ## **General Response**
>
> We would like to begin by expressing our sincere gratitude for the depth and rigor of your review. It is increasingly rare to receive such detailed feedback that reflects a deep, manual engagement with the manuscript, and we have the utmost respect for the time and effort you invested in analyzing our work. We are also encouraged by your specific ratings (Soundness: 2, Presentation: 3, Contribution: 3); this positive recognition has significantly boosted our motivation to further refine and improve this research.
>
> In response to your comments, we have revised the manuscript to clarify that our core novelty lies in a standardized output tokenization protocol across five categories of tasks (Figure 3), include ESM-2 + Linear Probe baselines across feasible tasks, verified the supervised nature of our ProteinGym benchmark to address data leakage concerns, and completely rewrote the Discussion section to transparently analyze the limitations of the VQ-VAE tokenizer and the specific engineering choices required for multi-task learning.
>
> We apologize for the extent of this detailed response in the following, but we are driven by a strong conviction in the significant value of this work; we believe it warrants this comprehensive discussion to clear up existing misconceptions and fully illuminate the potential of our tokenization framework.

---

> ### Author Response · Authors · 2025-11-24
>
> ## **Response to Evaluation of ProteinGym**
>
> We addressed the reviewer's concern regarding the magnitude of our performance improvement. We believe this concern likely stems from a comparison with the **Zero-Shot** benchmarks often reported in the literature, whereas our method utilizes the **Supervised** version of the ProteinGym dataset, which is less reported in the literature.
>
> Since our approach is inherently supervised, comparing it against zero-shot baselines would be inappropriate. We strictly adhered to the cross-validation segments, indices, and labels defined by the dataset's official supervised benchmark protocol. Following your comment, we rigorously double-checked our data splits and indices and confirmed that there were no errors in our implementation. Furthermore, we have now included the ESM-2 baseline in the updated manuscript, which shows performance alignment with Prot2Token. This suggests that if any data leakage exists, it would be inherent to the dataset's provided cross-validation splits themselves, rather than our specific methodology.
>
> We are fully committed to the rigor of this work and are open to moving this result to the appendix should you view it as an outlier that requires a more nuanced discussion.
>
> ## **Response to Concerns on Weak Baselines**
>
> We have updated the manuscript to include an **ESM-2 + Linear Probe** baseline to ensure a rigorous comparison. Our rationale is as follows:
>
> Our goal was not to claim state-of-the-art (SOTA) performance against methods utilizing highly specialized structure-based encoders or auxiliary features. Instead, we aimed to isolate the effectiveness of our generative decoder approach. Therefore, Prot2Token (ESM-2 + Causal Decoder) must be compared against a fair baseline: ESM-2 + Specialized Probe for most tasks. We have updated our tables to include this comparison in all feasible tasks to demonstrate that a Causal Decoder can function effectively as a task-agnostic predictor. This allows us to highlight the specific value of the generative head while keeping the backbone encoder identical.
>
> ## **Response to Concerns on Ligand Binding Formulation**
>
> We clarify that our objective in this specific experiment was not to build a zero-shot protein-ligand binding site predictor for unseen chemical compounds, but rather to demonstrate the efficacy of treating distinct prediction targets as unique task tokens. We justify this design choice based on three key factors:
>
> **1. Practicality and Baselines:**
> Utilizing a fixed set of supported ligands remains highly practical for many applications, particularly for metal ions which have a naturally predefined range. We specifically compared our method against recent protein-ion prediction methods in **Table 26** of the appendix, as few existing methods support more than 10 distinct ligands/ions simultaneously.
>
> **2. Interpretability of Task Tokens:**
> A primary motivation for using discrete task tokens was to enable deep interpretability of the learned representations (Appendix A.4.4). This setup allowed us to compare ligand representations directly, a form of analysis that would be difficult with continuous chemical embeddings:
>
> > In Appendix A.5, we investigated the interpretability of our learned task tokens for the protein-ligand binding site task by analyzing the embeddings of 41 unique ligand tokens. We performed dimensionality reduction and clustering using PCA and K-Means, finding that the learned token embeddings formed clusters strongly correlating with the ligands' physicochemical properties—such as molecular weight and net charge (ARI = 0.447). This semantic organization in the token space highlighted a crucial multi-task learning effect: by effectively sharing information across task tokens, we observed that the model could leverage knowledge from well-represented ligands to significantly boost the prediction accuracy for underrepresented (tail) ligands. This finding validates the synergy inherent in our unified approach, demonstrating that low-resource tasks benefit directly from the shared latent representations learned from high-resource tasks.
>
> **3. Comparison with Chemical Encoders:**
> We empirically explored the alternative approach suggested by the reviewer: using a chemical encoder to represent ligands via SMILES strings. However, we found that performance using current pre-trained chemical encoders was inferior to our task-token approach. This observation aligns with findings in recent literature, such as *"Zero-Shot Protein–Ligand Binding-Residue Prediction from Sequence and SMILES"*, which suggests that current chemical encoders may not yet be as effective as protein encoders in this specific binding site context.
>
> We have updated the appendix of the manuscript to include this justification in Section A.4.4 of the appendix explicitly.

---

> ### Author Response · Authors · 2025-11-24
>
> ## **Response to Concerns on Cross-Task Learning Evaluation**
>
> We acknowledge that a systematic "leave-one-out" analysis or a full task influence matrix would provide a comprehensive view of negative transfer. However, conducting such an exhaustive evaluation across all task permutations was computationally prohibitive given our resource constraints (4x A100 GPUs).
>
> Furthermore, our selection of task combinations was not arbitrary but rather strategically driven by the need to address severe data heterogeneity. We observed that several protein-prediction tasks exhibit extremely imbalanced label distributions, making joint training non-trivial compared to other fields. For instance:
> * **Classification:** As detailed in Appendix Section A.4.1, more than a dozen classes in the Fold classification task contain only a single sample in the training or test sets. This creates significant training instability, which we successfully mitigated by carefully selecting auxiliary tasks to provide regularization.
> * **Binding Site Prediction:** Similarly, in Appendix A.4.3, numerous site index tokens are heavily under-represented compared to dominant index tokens. We addressed this by introducing self-supervised pre-training of the decoder.
>
> Jointly training all such tasks simultaneously would require extensive engineering efforts to balance these specific data irregularities (loss weighting, sampling strategies) rather than yielding clear scientific insight into the architecture itself. Consequently, we observed a strong correlation between the number of samples (Table 11) and training stability. In cases with sufficient data, Prot2Token trained stably and consistently outperformed the **ESM-2 + Linear Probe** baseline. Therefore, we attribute the challenges in scaling to "all-task" training to these specific data limitations rather than a fundamental limitation of the Prot2Token architecture.
>
> ## **Response to Concerns on Binding Site Prediction and Sorting Order**
>
> In following, we respectfully address the concern that predicting binding sites via sorted indices is "fragile." Based on our architectural design and supporting literature, we argue that the specific permutation of the target indices does not negatively impact performance. In other words, the provided ordering strategy is predictable because it has a deterministic format.
>
> * **1. Bidirectional Encoder Context:**
> The critical factor here is that the decoder is conditioned on the embedding provided by the **ESM-2 encoder**, which is bidirectional. This means that at every step of generation, the decoder has full attention access to the entire protein sequence context simultaneously. Whether the model is asked to generate indices from low-to-high (left-to-right) or high-to-low, the information availability regarding specific residues remains identical. The model simply learns the "sorting" as a deterministic syntax rule for the output.
>
> * **2. Support from Literature (Pix2Seq):**
> This robustness to ordering in autoregressive set prediction has been well-documented in other domains. For example, the **Pix2Seq** framework (*Pix2Seq: A Language Modeling Framework for Object Detection*, Chen et al.) explicitly explored this in **Figure 4**, demonstrating that autoregressive models can effectively learn to generate sets of objects (bounding boxes) regardless of the specific ordering strategy, as long as the training target has a consistent and deterministic format. Our approach aligns with these findings: the "sorted index" strategy is not a fragility, but rather a standard method for both binding site and self-supervised tasks, serializing a set into a sequence for autoregressive modeling.

---

> ### Author Response · Authors · 2025-11-24
>
> ## **Response to Concerns on Regression Tokenization (ASCII & Directionality)**
>
> We thank the reviewer for highlighting this critical aspect of our methodology. We argue that the standard left-to-right tokenization of continuous values is not merely an arbitrary choice, but a mechanism that introduces a beneficial **coarse-to-fine inductive bias**.
>
> * **1. Hierarchical Prediction as Regularization:**
> By tokenizing numbers digit-by-digit from most significant to least significant, we force the model to first predict a rough estimation (coarse value) before refining the prediction with subsequent tokens (fine precision). This hierarchical structure acts as a strong regularizer. Unlike a standard regression head, which predicts a value in a single "shot," our approach effectively gives the model multiple opportunities to attend to the context and refine its scalar prediction. We believe this contributes to the robustness against overfitting that we observed in our experiments.
>
> * **2. Validation via Right-to-Left Ablation:**
> We did empirically test the reverse order (Right-to-Left tokenization) beforehand. As hypothesized, this destroyed the coarse-to-fine bias, forcing the model to predict least-significant digits without an established context for the magnitude. This resulted in measurable performance degradation compared to the standard order. Given that Prot2Token consistently outperforms the **ESM-2 + Linear Probe** baseline in most of the regression tasks with left-to-right as our standard tokenization of the regression task. We have updated the manuscript to add this explanation in **Appendix A.2.2**.
>
> * **3. Normalization Strategy:**
> We further standardized this process by min-max scaling labels to the [0, 1] range with fixed four-digit precision (as detailed in **Appendix A.4.2**). We found that this consistent range allowed the decoder to learn the structural syntax of numerical outputs significantly faster than unscaled values.
>
> ## **Response to Concerns on Architectural Choices (Unfrozen Layers)**
>
> We thank the reviewer for highlighting this hyperparameter choice. We clarify that the decision to unfreeze different numbers of encoder layers was not arbitrary, but rather dictated by the inherent heterogeneity in task difficulty ("hardness") and the required model capacity for distinct biological problems.
>
> Different tasks necessitate varying degrees of adaptation from the pre-trained ESM-2 backbone. For instance, predicting **Fluorescence** intensity relies on distinct structural features compared to predicting **Human Protein-Protein Interactions (PPI)**, which involves complex interfacial dynamics. Unfreezing a static number of layers across all tasks typically leads to suboptimal results—either underfitting on complex tasks or overfitting on simpler, data-poor tasks. We have explicitly added this justification to **Section 3 (Experiments)** of the updated manuscript to clarify how task complexity guided these architectural decisions.
>
> ## **Response to Concerns on Structure Prediction Quality and VQ-VAE**
>
> We thank the reviewer for this insightful suggestion, which aligns perfectly with the direction of our future research. Regarding the performance gap, it is important to contextualize our results against the **no-MSA version of AlphaFold2**. In this comparison, Prot2Token achieves slightly superior performance while offering a dramatic efficiency advantage, approximately **54x faster** inference speed. We believe this speed-accuracy trade-off is valuable for specific applications where generating MSAs is the primary bottleneck.
>
> Regarding the absolute quality gap compared to full AlphaFold2, we agree that the VQ-VAE is the primary bottleneck. At the start of this project, we selected the most robust fully open-source 3D VQ-VAE available at the time, relying on the reconstruction metrics reported in its original publication. However, after our training, we observed a performance plateau and subsequently discovered that the model’s actual reconstruction error on our test sets was significantly higher than initially reported. Unfortunately, given that 3D structure prediction training is substantially more computationally intensive than other tasks, iterating on alternative VQ-VAE architectures was not feasible within our limited computational timeframe. We strongly considered switching to the ESM-3 tokenizer; however, our access to GPU resources was curtailed before we could initiate that training run. We are currently applying for additional compute resources to specifically address this. We remain confident that integrating a higher-fidelity VQ-VAE (such as ESM-3's) will significantly close the quality gap in the next iteration of this framework, as no sign of convergence we saw in Figure 15.

---

> ### Author Response · Authors · 2025-11-24
>
> ## **Response to Comment on Missing Principled Baselines**
>
> We appreciate this comment and acknowledge that we did not compare against the absolute state-of-the-art (SOTA) for every individual task. This was a decision based on the specific scope of our contribution.
>
> Many current SOTA methods achieve their performance by incorporating additional modalities (such as experimental 3D structures or MSAs) or by engineering highly specialized, task-specific architectures. Since Prot2Token relies solely on the protein sequence and a unified decoder, comparing it against such specialized systems would conflate the benefits of those auxiliary inputs with the architectural differences we aimed to study.
>
> Instead, our goal was to isolate the effectiveness of the **Generative Causal Decoder** compared to standard predictive heads. Therefore, the most "principled" baseline for our specific claim is **ESM-2 + Specialized Head** (e.g., Linear Probe or standard regressor) versus **ESM-2 + Causal Decoder** (Prot2Token). This comparison strictly measures how close a generalist, next-token prediction framework can get to a specialized specialist while using identical input information. As mentioned in our responses to other reviewers, we have updated the manuscript to explicitly include these **ESM-2 + Specialized Probe** baselines across all feasible tasks, ensuring a fair and rigorous evaluation of our proposed paradigm.
>
>
> ## **Response to Skepticism on the "Everything is Auto-Regressive" Philosophy**
>
> We understand the reviewer's hesitation regarding such a maximalist approach. However, we respectfully argue that the value of this work lies precisely in establishing the *foundational protocol* for this philosophy. Our primary goal was to create a robust tokenization strategy that maps the diverse landscape of protein-prediction tasks into five distinct archetypes (detailed in **Appendix A.2**). We demonstrated that from a machine learning perspective, virtually all protein tasks can be mapped to one of these formats.
>
> To better address this balance of promise vs. complexity, we have completely rewritten the **Discussion** section to more transparently reflect the achievements, current limitations, and future trajectory of this framework. Specifically, Prot2Token provides several distinct advantages that specialized heads cannot:
>
> * **Universal Protocol as a Guideline:** We established a unified guideline for these five task categories. For tasks with well-distributed labels, Prot2Token is capable of effective prediction without architectural changes. For complex and disparate tasks like **3D structure prediction** and **kinase-specific phosphorylation site prediction**, the autoregressive approach offers significant advantages in inference speed (e.g., ~1000 faster for structure) and output flexibility. By converting them into a causal language modeling format, we unify them at the prediction level, which is unique only to our work.
>
> * **PLM-Agnostic Adaptation:** Prot2Token decouples the prediction head from the backbone, allowing us to leverage the representation capability of *any* pre-trained protein language model (PLM) via a simple autoregressive predictor. This avoids the need to design custom heads for every new PLM released.
>
> * **Superiority in Regression:** As discussed in previous points, our approach consistently outperformed the linear probe baseline for regression tasks. This confirms that the **coarse-to-fine regularization** inherent in our tokenization strategy provides a genuine modeling advantage over standard one-shot regression.
>
> * **Potential for Multi-Task Integration:** While we have only scratched the surface, Prot2Token provides the necessary scaffold to merge disparate tasks into a single multi-task learning run, something impossible with heterogeneous prediction heads.
>
> * **Bridge to General-Purpose LLMs:** Perhaps most importantly, this framework provides a pathway to upgrade General Purpose LLMs. By converting biological tasks into a language-compatible format, we pave the way for future models that can seamlessly alternate between scientific reasoning and direct protein property prediction, speeding up scientific discovery.

---

### Official Review · Reviewer_pn8H · 2025-10-31

**Soundness:** 3
**Presentation:** 3
**Contribution:** 2
**Rating:** 4
**Confidence:** 4

**Summary:**

This paper introduces Prot2Token framework which tries to handle various protein tasks using a unified decoder, including classification, regression, binding-site prediction, and even sequence-to-structure generation.
The key idea is to turn every task into a next-token prediction problem, all targets are serialized into tokens, and each task is triggered by a learned “task token”.
The framework uses ESM2 as the protein encoder, optionally fused with a SMILES encoder.
Experiments across benchmarks like DeepLoc, ProteinGym, and CAMEO show reasonable performance and strong throughput.

**Strengths:**

1. this paper addresses a challenging issue in the downstream application of PLM: additional head across each tasks and architectures
2. this paper leverages a straightforward idea that turns everything into a next-token prediction framework training
3. the experiments spans multiple benchmarks including both cls/reg-level and sequence-level
4. paper writing is clear about its goals and experimental setup.

**Weaknesses:**

My concern mainly falls in the conceptual depth of "unified decoding" proposed in this paper, with several minor issues.
1. The "unified decoding" claimed in this paper mostly coms from designing a token vocabulary for task tokens. Functionally, this isn’t very different from instruction- or prefix-based tuning, it’s essentially another form of prompt learning in the view of learnable tokens. While the paper criticizes prior works for relying on prompt engineering, it ends up doing the same thing in a more structured way. The unification happens at the tokenization level, not at a modeling or representation level.
2. Some of the comparison models don’t use ESM-2 as their encoder. Since the proposed method builds on ESM-2, including ESM-2-based baselines would make the comparison fairer and the paper more convincing.
3. The description of the self-supervised warm-up and multi-task training is a bit confusing. It’s not clear whether the warm-up applies only to the binding-site prediction task, or whether all tasks benefit from it. Likewise, the paper doesn’t clearly state if the pretraining and multi-task learning are done in one continuous run or in separate stages.

**Questions:**

All my concerns about this paper are listed in "Weaknesses" section, please refer to the weakness part for rebuttal/discussion.

---

> ### Author Response · Authors · 2025-11-24
>
> # Summary
>
> We sincerely thank the reviewer for their positive assessment of the soundness and presentation of our work, and for the insightful critique regarding the conceptual depth of our unified decoding approach. We appreciate your constructive feedback, which has helped us refine the positioning of our contribution. In response to your comments, we have revised the manuscript to clarify that our core novelty lies in a standardized output tokenization protocol across five categories of tasks (Figure 3), rather than merely input prompting. We have also incorporated ESM-2 based baselines to ensure fairer comparisons and explicitly restructured the experiment section to clearly distinguish the self-supervised pre-training stage from the subsequent multi-task fine-tuning phase.

---

> ### Author Response · Authors · 2025-11-24
>
> ## **Response to Concern 1:**
>
> We appreciate this insightful critique regarding the conceptual depth of our approach. We respectfully wish to clarify that our definition of "unified decoding" extends beyond the use of learnable task tokens at the input level.
>
> **1. Standardization Protocol vs. Architectural Engineering:**
> As the reviewer noted, we utilize task tokens, but these are primarily a mechanism to enable multi-tasking within a single training session rather than the central innovation of the work. Our core novelty lies in creating a standardized protocol that converts disparate biological tasks into a uniform sequence of tokens. Rather than designing specialized heads or architectures for each task (which is the current standard in the field), we demonstrated that these five task groups can be solved using a single, unified decoder architecture (detailed in **Appendix A.2 Tokenization** and updated **Figure 3**): Classification, Regression, Binding Site, Sequence-to-Sequence, and Other. This shifts the focus from architectural engineering to tokenization strategy, bringing a broad range of protein tasks into a generative AI interface.
>
> **2. Novel Output Tokenization (Updated Figure 3):**
> While instruction tuning focuses on the *input*, our unifying approach focuses heavily on the *output*. As illustrated in the updated **Figure 3**, we developed novel tokenization strategies for complex, structured outputs such as **Post-Translational Modifications (PTM)** (Figure 3B) and **Binding Site Indexing** (Figure 3C). These tasks traditionally require highly specialized regression or pointer-network architectures. By converting them into a causal language modeling format, we unify them at the representation level.
>
> **3. Modeling Implications (Self-Supervised Pre-training):**
> This tokenization strategy is not merely a surface-level formatting change; it unlocks new modeling capabilities. Specifically, transforming continuous or index-based labels into a discrete vocabulary allowed us to introduce a **self-supervised pre-training objective for the decoder**. This method addresses the label sparsity and imbalance inherent in binding site and PTM tasks—an algorithmic innovation that would be impossible with standard regression heads or simple prefix-tuning.
>
>
> ## **Response to Concern 2**
>
> We agree that controlling for the encoder is essential for a fair assessment. We have included ESM-2-based baselines in our comparisons for the majority of tasks where a straightforward baseline comparison was feasible. The manuscript has been revised for this update.
>
> ## **Response to Concern 3**
>
> We agree with the reviewer that the distinction between these stages was not sufficiently clear in the main text.
>
> The self-supervised pre-training of the decoder is performed as a **separate, initial step**. Its purpose is to create a pre-trained decoder which is subsequently used to initialize the model for specific tasks (specifically Binding Site and PTM prediction), replacing standard random initialization. As noted in Appendix A.4.3: *"We introduced a self-supervised pre-training strategy to effectively embed inductive biases into the decoder before fine-tuning it on the main supervised tasks."*
>
> We have resolved this ambiguity by explicitly detailing this two-stage process in **Section 3.3** of the main body. The text now clearly distinguishes the pre-training phase from the subsequent supervised training phase to avoid confusion.

---

### Official Review · Reviewer_11VW · 2025-11-02

**Soundness:** 2
**Presentation:** 2
**Contribution:** 1
**Rating:** 2
**Confidence:** 4

**Summary:**

The paper proposes Prot2Token, a unified framework that reformulates various protein-related tasks—such as classification, regression, and structure prediction—into a next-token prediction problem. The authors combine pre-trained protein encoders (mainly ESM2) with an autoregressive decoder conditioned on task tokens. The idea is to create a single model capable of handling diverse prediction formats through a shared tokenization and decoding process.

While the paper claims to present a unifying paradigm similar to GPT-style modeling in NLP, the method mainly reuses existing building blocks (ESM encoder, cross-attention fusion, VQ-VAE structure tokenizer) without introducing substantial architectural or algorithmic novelty.

**Strengths:**

- A good motivation: to build a unified model for every different protein tasks

- Reasonable engineering effort to connect multiple existing components: for example, they unify different downstream tasks with one model.

- Consistent writing: the writing is clear

**Weaknesses:**

- **Claim is big, but paper is unable to support the claim**: the biggest issue is that, this paper claims that they try to advance a huge step in the protein field, by unifying different tasks into one model, with some prompt tokens. However, the method part significantly lacks  novelty. It just stated what they have used for model module building. Just stacking together without any deeper insights (either theoretical or empirical). Even worse, the performance tables show very limited baselines (some definitely out-of-dated). For example, table 3 shows the performance in PEER's benchmarking paper from 2023, which is already beaten by a lot of methods later on.

- **Illustration figures look like workshop quality instead of conference quality**: many specific details without high-level deep insights.

**Questions:**

Please see the weakness. Sincerely suggest to incorporate much more comprehensive benchmarking results (baselines, metrics) to support your strong and big claim, to make it empirically solid, given that this is more like an application driven paper.

---

> ### Author Response · Authors · 2025-11-24
>
> # Summary
>
> We sincerely thank the reviewer for their critical assessment and for highlighting areas where the manuscript required greater empirical rigor and conceptual clarity. We have taken your feedback seriously and revised the paper. Key updates include the addition of ESM-2 + Linear Probe baselines across more tasks to strictly isolate the benefits of the generative decoder. We also refined the narrative to explicitly frame Prot2Token as a unified tokenization protocol that maps tasks into five categories, rather than claiming a monolithic universal model. Furthermore, the Discussion section has been completely rewritten to provide a deeper analysis of key insights, limitations, and future directions.

---

> ### Author Response · Authors · 2025-11-24
>
> ## **Response to Concerns regarding Novelty and Baselines**
>
> We thank the reviewer for their critical assessment. We respectfully wish to clarify that the primary contribution and novelty of this work is not simply "stacking" existing modules, but rather the development of a unified tokenization protocol that effectively maps the vast majority of protein-prediction tasks into a cohesive generative framework (Figure 3). To our knowledge, this is the first attempt to categorize diverse protein tasks into five distinct archetypes—Classification, Regression, Binding Site, Sequence-to-Sequence, and Other—and reformulate them entirely into a next-token prediction objective standard in current generative AI.
>
> To address your specific concerns regarding novelty, baselines, and empirical depth, we offer the following clarifications:
>
> * **Unified Tokenization Protocol:** Our novelty lies in creating a standardized protocol that converts disparate biological tasks into a sequence of tokens. Rather than designing specialized heads or architectures for each task (which is the current standard in the field), we demonstrated that these five task groups can be solved using a single, unified decoder architecture (detailed in Appendix A.2 Tokenization): Classification, Regression, Binding Site, Sequence-to-Sequence, and Other. This shifts the focus from architectural engineering to tokenization strategy, bringing a broad range of protein tasks into the generative AI paradigm.
>
> * **Appropriate Baseline Comparisons:** Our goal was not to claim SOTA performance against methods utilizing highly specialized structure-based encoders or auxiliary features. Instead, we aimed to isolate the effectiveness of our generative decoder approach. Therefore, Prot2Token (ESM-2 + Causal Decoder) must be compared against a fair baseline: ESM-2 + Specialized Probe for most tasks. We have updated our tables to include this comparison in all feasible tasks to reflect that a Causal Decoder can be a task-agnostic predictor. This demonstrates the value of the generative head over a standard linear head while keeping the backbone the same. We would appreciate it if the reviewer could suggest other baselines that utilize a frozen ESM-2 backbone without modifying the encoder, as these would provide the most fair comparison.
>
> * **Deep Empirical Analysis (Multi-Task Learning):** Contrary to the view that the paper lacks deep insights, we provided a rigorous analysis of multi-task learning in the context of protein-ligand binding sites. As detailed in **Table 25** and **Appendix A.5 (Protein-Ligand Binding Site Task Tokens Interpretation)**, we merged 41 distinct ligand prediction tasks into a single training session by inventing a self-supervised pre-training.
>
> * **Interpretability of Task Tokens:** Our analysis went beyond performance metrics. We analyzed the learned task token embeddings and discovered strong correlations between the embedding clusters and the physicochemical properties of the ligands. We measured how this semantic relationship improved predictions for underrepresented samples (tail distribution), validating that the model is learning meaningful chemical relationships rather than just memorizing labels.
>
> * **Computational Efficiency in Complex Tasks:** Finally, we emphasize the results regarding 3D structure prediction. The significant inference speed advantage observed in Prot2Token serves as a strong signal for the utility of the next-token prediction format, particularly for tasks that traditionally require extensive specialized computation.

---

> ### Author Response · Authors · 2025-11-24
>
> ## **Response to Comment on Figure Quality**
>
> We would appreciate further clarification on this point, as we have previously received positive feedback regarding the level of abstraction and clarity in our figures. Our visual strategy was intentional, aiming to balance conceptual overview with necessary technical reproducibility. To this end, we structured our figures into two distinct categories:
> * **High-Level Abstraction:** Figures 1, 3, and 11 are designed to convey the broad architectural concepts and workflow without overwhelming the reader with minutiae.
> * **Methodological Detail:** Figures 2, 4, 5, and 8 incorporate granular details specifically to assist in understanding the tokenization logic and model mechanics.
>
> Furthermore, to ensure high standards of visual communication, we engaged a professional UI/UX designer to establish a unified design system for the paper. This ensures consistency in iconography, color schemes, and layout across all method-related figures. Our design philosophy was specifically tailored to bridge the gap between our two core audiences: providing necessary structural details for Computer Science researchers while maintaining accessible workflows for the Bioinformatics community.
>
> Could you please provide more specific details regarding which elements appeared lacking in depth or polish? We are eager to understand your perspective to ensure we avoid similar pitfalls in our future work and to improve the current manuscript where possible.

---

### Official Review · Reviewer_wnsV · 2025-11-03

**Soundness:** 2
**Presentation:** 3
**Contribution:** 2
**Rating:** 6
**Confidence:** 3

**Summary:**

This paper presents Prot2Token, a method for unifying multiple downstream classification tasks using an autoregressive decoder. A pre-trained protein language model (PLM) is used to generate representations of an input protein sequence, which is then fused with ligand representations from a small molecule foundation model, if applicable, and fed to an autoregressive decoder that generates tokenized outputs specific to a given task. A task token is also learned and is fed as a precursor to the decoder to make it aware of the task under study. Numerical results demonstrate that Prot2Token outperforms other baselines across various benchmarks.

**Strengths:**

- Representing diverse downstream classification and regression tasks for protein sequences using a universal tokenization scheme is, in my opinion, interesting and novel.
- Performance improvements over prior methods is significant in some benchmarks.

**Weaknesses:**

1. To me, the very shortcoming that the paper is trying to address is its main weakness. As the authors allude to, a universal model that can support any given downstream task is computationally prohibitive. Therefore, the method has been limited to a few downstream tasks only.
2. Following (1), it seems that different versions of Prot2Token were trained on different (combinations of) downstream tasks (Prot2Token-A/B/C/D). Is my understanding correct? If so, it defeats the purpose of having a universal model. If not, what do the auxiliary tasks in Tables 2 and 7 imply, and how are they selected?
3. Comparisons with baselines are slightly unconvincing to me. At the very least, I would suggest including the results of a frozen ESM-2 model, combined with a separate linear probe for every task.
4. Following (1) and (3), the main novelty of the approach is the universal decoding engine, which tries to overcome the alternative, which is separate, heterogeneous downstream prediction heads specific to each task. I wonder if having multi-layer downstream prediction heads that share the first few layers and differ only in their last layer would also reap the same benefits as multi-task learning, as seen in your method.

**Questions:**

- When including auxiliary tasks (e.g., in Tables 2 and 7), does the performance on those auxiliary tasks improve, too? Or is the improvement only observed in the target task?
- Could you please provide more details on the task tokens and whether they could generalize to a task unseen during training (e.g., in a meta-learning way)? What is $m$ on lines 208-210?

---

> ### Author Response · Authors · 2025-11-24
>
> # **Summary:**
>
> We sincerely thank the reviewer for their detailed evaluation and for acknowledging the novelty of our universal tokenization scheme. We appreciate your positive assessment of the presentation (3) and have taken your ratings on soundness (2) and contribution (2) as a strong motivation to strengthen the empirical rigor of our work.
>
> Changes: We have extensively revised the manuscript to strengthen empirical rigor and conceptual clarity. Key updates include the addition of ESM-2 + Linear Probe baselines across feasible tasks to strictly isolate the benefits of the generative decoder. We also refined the narrative to explicitly frame Prot2Token as a unified tokenization protocol that maps tasks into five categories, rather than claiming a monolithic universal model. Furthermore, the Discussion section has been completely rewritten to provide a better analysis of the insights, limitations and future work.

---

> ### Author Response · Authors · 2025-11-24
>
> # **Answers to weakness:**
> ### **Response to Weakness 1**
>
> We appreciate this thoughtful comment. While we agree that a universal model involves significant computational overhead, this cost is incurred primarily during the training phase. Consequently, this upfront investment results in highly efficient inference. Given our limited available computational resources (4x A100 GPUs), we strategically chose to demonstrate the feasibility of our tokenization approach across distinct **types** of protein-prediction tasks, rather than maximizing the sheer number of tasks. To this end, we categorized the majority of protein tasks into five structural formats (detailed in Appendix A.2 Tokenization): Classification, Regression, Binding Site, Sequence-to-Sequence, and Other.
>
> From a machine learning perspective, tasks within each category can be treated with the same tokenization protocol despite their underlying biological differences. We focused on representing at least one task per category for Prot2Token. To demonstrate that this approach generalizes to other tasks within these categories, we included additional experiments in the appendix. For instance, regarding regression tasks, we tested stability, protein thermostability, and protein-protein binding affinity (Appendix Section A.4.2 Regression). This consistency holds for other task types as well. Our rationale was to explore at least one candidate for each category to demonstrate the task-agnostic nature of the tokenization via next-token prediction, rather than overwhelming the experiments and inflating computational costs.
>
> Furthermore, we observed that several protein-prediction tasks exhibit extremely imbalanced label distributions. For instance, as detailed in Appendix Section A.4.1 (CLASSIFICATION), more than a dozen classes in the Fold classification task contain only a single sample in the training or test sets. This creates training instability which we mitigated by adding auxiliary tasks. Similarly, in binding site prediction (Appendix A.4.3 SELF-SUPERVISED PRE-TRAINING OF DECODER), numerous site index tokens are heavily under-represented compared to dominant index tokens. We addressed this issue by introducing self-supervised pre-training of the decoder. Jointly training such tasks alongside others would require extensive engineering efforts to balance these specific data irregularities rather than yielding scientific insight into the architecture itself. Consequently, we observed a strong correlation between the number of samples (Table 11) and training stability. In cases with sufficient data, Prot2Token trained stably and consistently outperformed the ESM-2 encoder + linear prob baseline. Therefore, we attribute these specific challenges to data limitations rather than a limitation of the Prot2Token architecture.
>
>
> ### **Response to Weakness 2**
>
> Our primary novelty is the unified tokenization protocol that standardizes diverse protein tasks, rather than a single monolithic model. Due to GPU constraints, we utilized varied configurations (Prot2Token A–D) to align model capacity with task complexity; simple tasks incur unnecessary computational costs with large decoders, whereas complex tasks like 3D structure prediction necessitate them. We demonstrated the efficacy of this multi-task framework in protein-ligand binding (Table 25), where increasing the number of unique task tokens consistently improved performance, confirming effective information sharing.
>
> - **How are the auxiliary tasks selected?**
>
> Given the extensive search space for potential auxiliary tasks, we filtered the candidates for Tables 2 and 7 based on two primary factors to ensure compatibility with the target task:
> 1. Similarity in the number of training samples.
> 2. Comparable convergence speeds.
>
> ### **Response to Weakness 3**
>
> We appreciate this suggestion and agree that it strengthens the evaluation. We are currently updating our experimental results to include the requested baseline: a frozen ESM-2 model combined with a linear probe. These updated tables will ensure a direct and fair comparison between the baseline and Prot2Token.
>
> ### **Response to Weakness 4**
> We prioritized a homogeneous next-token objective to avoid the optimization complexity inherent in heterogeneous heads, which require balancing distinct loss functions, value ranges, and convergence rates. Our unified tokenization protocol circumvents this by enabling a single, shared cross-entropy loss. While we did not perform a direct ablation, established benchmarks like PEER have previously demonstrated the performance benefits of this multi-task approach, supporting our architectural design.

---

> ### Author Response · Authors · 2025-11-24
>
> # **Answers to questions:**
> ### **Response to Question 1**
>
> We did not observe any noticeable performance changes for the auxiliary tasks listed in Tables 2 and 7. We noted this point in the
>
> ### **Response to Question 2**
>
> We appreciate this thoughtful comment. We have reported an extensive analysis of task token embeddings for protein-ligand binding site prediction by treating 41 types of ligands as unique task tokens. We mention this at the end of Section 3.3 (BINDING SITE) in the main body. Furthermore, we extensively evaluated and reported on the task token embeddings in Appendix A.5 (PROTEIN-LIGAND BINDING SITE TASK TOKENS INTERPRETATION). The summary of that section is as follows:
>
> In Appendix A.5, we investigated the interpretability of our learned task tokens for the protein-ligand binding site task by analyzing the embeddings of 41 unique ligand tokens. We performed dimensionality reduction and clustering using PCA and K-Means, finding that the learned token embeddings formed clusters strongly correlating with the ligands' physicochemical properties—such as molecular weight and net charge (ARI = 0.447). This semantic organization in the token space highlighted a crucial multi-task learning effect: by effectively sharing information across task tokens, we observed that the model could leverage knowledge from well-represented ligands to significantly boost the prediction accuracy for underrepresented (tail) ligands. This finding validates the synergy inherent in our unified approach, demonstrating that low-resource tasks benefit directly from the shared latent representations learned from high-resource tasks.
>
> ### **Address Overall Concerns Regarding the Prot2Token Approach**
> To better address this balance of promise vs. complexity, we have completely rewritten the **Discussion** section to more transparently reflect the achievements, current limitations, and future trajectory of this framework. Specifically, Prot2Token provides several distinct advantages that specialized heads cannot:
>
> - **Universal Protocol as a Guideline:** We established a unified guideline for these five task categories. For tasks with well-distributed labels, Prot2Token is capable of effective prediction without architectural changes. For complex, structured tasks like **3D structure prediction** and **binding site prediction**, the autoregressive approach offers significant advantages in inference speed (e.g., ~1000x faster for structure) and output flexibility.
>
> - **PLM-Agnostic Adaptation:** Prot2Token decouples the prediction head from the backbone, allowing us to leverage the representation capability of *any* pre-trained protein language model (PLM) via a simple autoregressive predictor. This avoids the need to design custom heads for every new PLM released.
>
> - **Superiority in Regression:** As discussed in previous points, our approach consistently outperformed the linear probe baseline for regression tasks. This confirms that the **coarse-to-fine regularization** inherent in our tokenization strategy provides a genuine modeling advantage over standard one-shot regression.
>
> - **Potential for Multi-Task Integration:** While we have only scratched the surface, Prot2Token provides the necessary scaffold to merge disparate tasks into a single multi-task learning run, something impossible with heterogeneous prediction heads.
>
> - **Bridge to General-Purpose LLMs:** Perhaps most importantly, this framework provides a pathway to upgrade General Purpose LLMs. By converting biological tasks into a language-compatible format, we pave the way for future models that can seamlessly alternate between reasoning with human language as well as direct protein property prediction, speeding up scientific discovery.

---

> > ### Comment · Reviewer_wnsV · 2025-11-25
> >
> > Thank you for your response and for the additional results and explanations. After reading the other reviewers' feedback, I would like to maintain my marginally positive rating of the paper, as I think, despite its limitations, the presented approach could lead to follow-up research on protein representation learning.

---

### Comment · Area_Chair_7mRB · 2025-11-28
**Please kindly check the rebuttal**

Dear Reviewers,

Thanks for your effort in reviewing the manuscript. Now the authors have provided the rebuttal and it's highly recommended to take a look and give your feedback. Thanks.

AC.

---

### Meta-Review · Area_Chair_Cc84 · 2025-12-12

**Summary:**

The paper tackles a relevant and ambitious goal: unifying diverse protein prediction tasks (classification, regression, binding site prediction, sequence-to-structure, etc.) into a single next-token prediction framework built on top of pretrained PLMs. However, reviewers identify several critical concerns:
- C1. The paper makes a big claim, "everything is next token prediction", but does not provide sufficient evidence to support this claim. In particular, there is no systematic analysis of cross-task interactions or negative transfer that would justify such a unifying claim.
- C2. The proposed method largely combines existing components with relatively limited ablation over a large design space. It is difficult to extract clear modeling insights beyond "make everything autoregressive and tokenize appropriately".
- C3. The experimental scope is broad, but the baselines are dated or relatively weak. Task-specific SOTA baselines are missing for several tasks.
- C4. For structure prediction, performance still lags substantially behind ESMFold and AF2. The current quality is far from broadly usable.

Overall, the conceptual novelty is modest. More importantly, the proposed method appears to impose an autoregressive tokenization paradigm on all tasks without a deeper theoretical or empirical analysis of when this is beneficial or appropriate. For several tasks, the results remain proof-of-concept rather than practically compelling. In my view, this work is promising but not yet fully mature in terms of conceptual clarity and evaluation depth for ICLR.

**Reviewer Concerns:**

The rebuttal addresses several concerns:
- The authors add ESM2 + linear/specialized probe baselines where feasible.
- They reframe the contribution, positioning the method as a unified tokenization protocol rather than a single universal model.
- They clarify several aspects of the methodology, task formulations, and the ProteinGym evaluation setup.

However, the most critical concerns listed above, especially regarding the strength of the central claim, the depth of conceptual insight, and the breadth/strength of baselines, remain outstanding.

**Reviewer Scores:**

Based on authors' responses, I expect most reviewers would keep their original scores (or change only marginally), as the rebuttal does not convincingly resolve the major issues around conceptual clarity and evaluation depth.

---

### Decision · Program_Chairs · 2026-01-26

Reject